# Buffer Matters: Unleashing the Power of Off-Policy Reinforcement Learning in Large Language Model Reasoning

**Xu Wan**♠ ♣ ♡, **Yansheng Wang**♣, **Wenqi Huang**◇, **Mingyang Sun**∗♡ ∗
♠ Zhejiang University ♣ Bytedance Seed Robotics ◇ China Southern Power Grid ♡ Peking University

## Abstract

Traditional on-policy Reinforcement Learning with Verifiable Rewards (RLVR) frameworks suffer from experience waste and reward homogeneity, which directly hinders learning efficiency on difficult samples during large language models post-training. In this paper, we introduce Batch Adaptation Policy Optimization (BAPO), an off-policy RLVR framework to improve the data efficiency in large language models post-training. It dynamically selects training batches by re-evaluating historically difficult samples and reusing high-quality ones, while holding a lower bound guarantee for policy improvement. Extensive experiments further demonstrate that BAPO achieves an average 12.5% improvement over GRPO across mathematics, planning, and visual reasoning tasks. Crucially, BAPO successfully resolves 40.7% of problems that base models consistently fail to solve. ○ *The code is available in* *Here*.

## 1 Introduction

Reinforcement Learning from Human Feedback (RLHF) has emerged as a transformative paradigm for aligning Large Language Models (LLMs) with human preferences and improving their performance on complex reasoning tasks (Ouyang et al., 2022; Bai et al., 2022). A significant recent evolution is Reinforcement Learning with Verifiable Rewards (RLVR) (Lambert et al., 2024), which replaces costly neural reward models with deterministic verification functions for more efficient and reliable training (Guo et al., 2025). Numerous on-policy RL optimization methods, particularly Group Relative Policy Optimization (GRPO) (Shao et al., 2024), and its variants like Dynamic Sampling Policy Optimization (DAPO) (Yu et al., 2025), Group Sequence Policy Optimization (GSPO) (Zheng et al., 2025), have demonstrated remarkable success in LLM post-training scenarios, achieving exceptional performance on mathematical reasoning, code generation, and various downstream applications (Yang et al., 2025; Chen et al., 2025a; Shen et al., 2025).

Although with lower bound guarantees of policy improvement theoretically (Mroueh, 2025), existing RL post-training frameworks still face significant efficiency challenges in practice. As shown in Figure 1, models after GRPO post-training struggle to handle difficult samples, especially those with zero accuracy in the initial rollout group. The reasons are twofold: (1) **Homogeneous rewards**: Recent investigations (Hong et al., 2025; Simoni et al., 2025) reveal that samples at both extremes of difficulty offer minimal benefit for post-training policy improvement. This arises because advantage estimation in most GRPO-based methods relies heavily on relative reward diversity within each group. Consequently, when intra-group rewards are identical, the lower bound guarantee for policy improvement col-

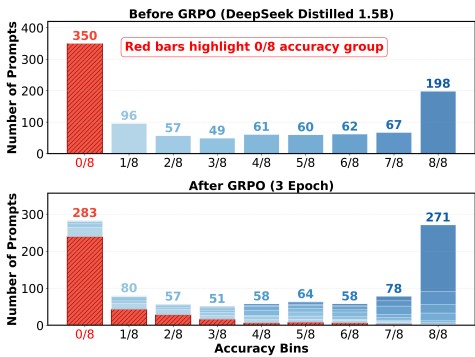

Figure 1: Tracking the sample counts across accuracy groups of the mathematical dataset before and after GRPO post-training.

---

∗Coresponding Author

lapses (Zhang et al., 2025; Mroueh et al., 2025), resulting in negligible effective gradient contributions (Liu et al., 2025; Yu et al., 2025). (2) **Waste of experience**: Given the sensitivity of policy improvement to intra-group reward variance, uneven difficulty distributions yield significantly fewer high-quality samples than the configured batch size implies. Crucially, since these methods are primarily on-policy and lack experience replay, each rollout group is consumed only once, leading to a substantial waste of valuable training data (Sun et al., 2025; Li et al., 2025).

A straightforward solution is to adopt off-policy rather than on-policy training paradigms, which has been established in traditional RL tasks as a viable solution to increase sample efficiency and diversity in the training batch (Queeney et al., 2021; Hilton et al., 2022; Meng et al., 2023). However, naively applying sample-reusing schemes to RL frameworks may exacerbate instability during LLM post-training, leading to entropy collapse, and ultimately performance degradation (Yu et al., 2025; He et al., 2025; Chen et al., 2025c).

Thus, to systematically exploring the utility of stale off-policy experience in RLVR post-training, we incorporates multiple off-policy strategies into on-policy RLVR framework to dissect effective pathways for historical data utilization. The main contributions of this paper are as follows:

(1) We propose a difficulty-aware experience replay mechanism as a practical solution for efficient off-policy data utilization. Unlike the simple mixing of the buffer's data and online data, we actively re-evaluate historical hard prompts to drive exploration while directly reusing high-quality trajectories with a dynamic quality threshold.

(2) Theoretically, we prove that under certain assumptions, the proposed adaptive construction mechanism mitigates the homogeneous reward issue via adaptive batch construction and KL-constrained updates.

(3) By integrating it into multiple reasoning tasks with different LLM backbones, we validate the proposed **B**atch **A**daptation **P**olicy **O**ptimization (BAPO) method achieves better convergence and yields greater improvements on solving difficult samples compared to existing on-policy and off-policy RLVR frameworks.

## 2 RELATED WORK

### 2.1 ON-POLICY RL POST-TRAINING FRAMEWORK

We first review the concept of on-policy RLVR, where the core objective is to optimize an LLM policy to maximize the outcome response reward. Let $x \in \mathcal{X}$ represent the input prompts, and $y \in \mathcal{Y}$ denote responses generated by the LLM policy $\pi_\theta$. The terminal reward $r(x, y) \in \{0, 1\}$ is determined by a deterministic verification function (Lambert et al., 2024; Guo et al., 2025). Following the setting of GRPO (Shao et al., 2024), the objective is formulated as:

$$\frac{1}{G} \sum_{i=1}^{G} \frac{1}{|y_i|} \sum_{t=1}^{|y_i|} \min\left(\rho_{i,t}(\theta)\hat{A}_{i,t}, \text{clip}(\rho_{i,t}(\theta), 1-\varepsilon, 1+\varepsilon)\hat{A}_{i,t}\right) - \beta \cdot \mathbb{D}_{\text{KL}}(\pi_\theta || \pi_{\text{ref}}) \quad (1)$$

where $\mathcal{G} = \{y_1, y_2, \dots, y_G\}$ represents a $G$-size group of responses sampled from $\pi_{\theta_t}(\cdot|x)$ for each input $x$; $\rho_{i,t}(\theta)$ is the probability ratio $\frac{\pi_\theta\left(y_i^t|y_i^{<t},x\right)}{\pi_{\theta_{\text{old}}}\left(y_i^t|y_i^{<t},x\right)}$ between current policy and old policy $\pi_{\theta_{\text{old}}}$ for the $i$-th responses' $t$-th token, $\varepsilon$ limits the magnitude of policy updates; and $\mathbb{D}_{\text{KL}}$ constrains the policy $\pi_\theta$ from deviating too far from a reference policy $\pi_{\text{ref}}$. Crucially, $\hat{A}_{i,t}$ denotes the estimated advantage of response $y_i$ for input $x$, which is derived from the standardization of rewards using the statistical properties of group $\mathcal{G}$. For the $i$-th response $y_i \in \mathcal{G}$ with reward $r_i = r(x, y_i)$, the estimated advantage is:

$$\hat{A}_{i,t} = \frac{r_i - \text{mean}(\{r_\ell\})}{\sqrt{\text{std}^2(\{r_\ell\}) + \varepsilon}} \quad (2)$$

where $\text{mean}(\{r_\ell\})$ and $\text{std}^2(\{r_\ell\})$ are the empirical mean and variance of rewards in group $\mathcal{G}$, respectively.

To enhance the practical efficiency of GRPO, a series of improved on-policy frameworks has been proposed. For instance, DAPO (Yu et al., 2025) sets distinct clipping ranges $\varepsilon_{\text{low}}$ and $\varepsilon_{\text{high}}$, and

employs a dynamic sampling strategy to ensure $\hat{A}_{i,t} \neq 0$. However, it consumes approximately four times the number of rollouts (Qu et al., 2025) compared to GRPO. Meanwhile, GSPO (Zheng et al., 2025) abandons the token-level ratio $\rho_{i,t}(\theta)$ and shifts to the sequence level $s_i(\theta)$, which has been validated to maintain more stable training, particularly in Mixture-of-Experts (MoE) architectures.

While the details of these methods vary, they all adhere to the on-policy framework for sampling and updates: the inference server is updated in synchronization with the trainer parameters, and the sampling strategy follows the " use-once-and-discard" principle throughout the training process.

## 2.2 OFF-POLICY RL POST-TRAINING FRAMEWORK

In contrast, as shown in Figure 2, off-policy RL post-training frameworks operate under a distinct paradigm, characterized by two core components: off-policy rollout for generating responses and off-policy training for constructing the training batch, as detailed below.

**Off-policy Rollout** avoids exclusive reliance on the current training policy for generation, instead leveraging past policies or external guidance. For example, AReaL (Fu et al., 2025) employs a fully asynchronous architecture that decouples generation from training, allowing rollout workers to use past policy. Mroueh et al. (Mroueh et al., 2025) fix the rollout policy on the vLLM inference server for multiple iterations to ensure stable sample generation. LUFFY (Yan et al., 2025) incorporates traces from stronger external policies to enhance reasoning capabilities beyond the model's initial limits.

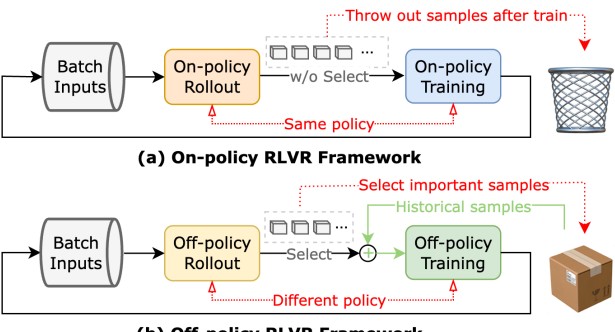

Figure 2: The overview of the (a) on-policy and (b) off-policy RL Post-training framework

**Off-policy Training** uses replay buffers to manage samples from historical policies with varying activation strategies. ARPO (Lu et al., 2025) dynamically samples non-zero reward samples from the buffer only when current batches contain all-zero rewards. DOTS (Sun et al., 2025) maintains a FIFO buffer that consistently reuses recent valid rollouts. RePO (Li et al., 2025) mixes buffer samples with on-policy samples using diverse retrieval strategies. ReMix (Liang et al., 2025) blends samples at fixed ratios while increasing the update-to-data ratio for efficiency. ReLIFT (Ma et al., 2025) stores high-quality solutions to challenging problems in its buffer and refines them through interleaved supervised fine-tuning. Kimi K1.5 (Team et al., 2025) stores both complete and partial trajectories to reduce temporal correlations while maintaining computational efficiency.

However, most off-policy RLVR methods ignore the policy stability of experiences. Samples entering the buffer at different training steps may exhibit varying policy distributions. These discrepancies introduce excessive noise into policy learning, which in turn exacerbates training instability. More importantly, simply reusing historical samples may even hinder the policy's improvement. The high-accuracy historical samples may cause the model to overly focus on existing reasoning paths with high advantages, suppressing the model's exploration capability and resulting in premature convergence to suboptimal solutions (Cui et al., 2025).

## 3 METHOD

In this section, we detail the core components of BAPO, particularly the adaptive construction strategy for the training batch, and provide a theoretical guarantee for the training stability of BAPO's policy update. Figure 3 provides an overview of the off-policy rollout and training workflow.

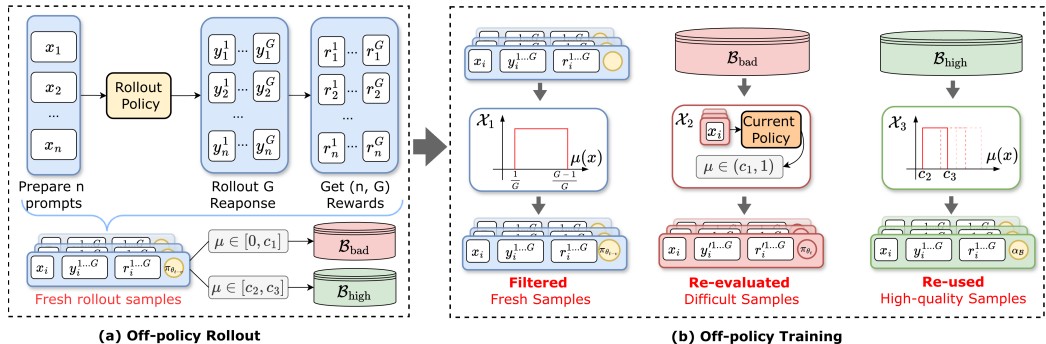

Figure 3: The workflow of (a) off-policy rollout and (b) off-policy training in our RLVR framework

## 3.1 FORMAL DEFINITIONS

We first formalize our training objective $\mathcal{L}_\alpha(\pi_\theta)$ as a combination of online rollout-derived and historical buffer-derived contributions:

$$\mathcal{L}_\alpha(\pi_\theta) = \underbrace{\mathbb{E}_{(x,y)\sim\alpha}\left[\rho_\alpha(\theta)\cdot\hat{A}(x,y)\right]}_{\text{Contribution from fresh samples}} + \underbrace{\mathbb{E}_{(x,y)\sim\mathcal{B}}\left[\rho_{\alpha_\mathcal{B}}(\theta)\cdot\hat{A}(x,y)\right]}_{\text{Contribution from historical samples}} - \beta\cdot\mathbb{D}_{\text{KL}}(\pi_\theta\|\alpha) \quad (3)$$

where $(x,y)\sim\alpha$ refers to filtered online samples from the rollout policy $\alpha = \pi_{\theta_{t-v}}$ with $v > 0$ representing the delay timesteps. $(x,y)\sim\mathcal{B}$ denotes historical samples from the replay buffer $\mathcal{B}$. The importance sampling ratios are defined as $\rho_\alpha = \frac{\pi_\theta(y|x)}{\alpha(y|x)}$ for the online rollout samples and $\rho_{\alpha_\mathcal{B}} = \frac{\pi_\theta(y|x)}{\alpha_\mathcal{B}(y|x)}$ for buffer samples, $\alpha_\mathcal{B}$ is the historical rollout policies that generated the buffer. Each entity in the buffer $\mathcal{B}$ is formally defined as:

$$\mathcal{B} = \{(u_i, \{x_{i,j}\}_{j=1}^G, \{y_{i,j}\}_{j=1}^G, \{r_{i,j}\}_{j=1}^G, \{\alpha_\mathcal{B}(y_{i,j}|x_i)\}_{j=1}^G)\}_{i=1}^{|\mathcal{B}|} \quad (4)$$

where $u_i$ is the unique identifier of each prompt, $\{x_{i,j}\}$, $\{y_{i,j}\}$, $\{r_{i,j}\}$ represent the set of prompts, generated responses, and corresponding rewards, respectively. $\{\alpha_\mathcal{B}(y_{i,j}|x_i)\}_{j=1}^G$ is the rollout policy's probability, which is stored for calculating $\rho_{\alpha_\mathcal{B}}(\theta)$ when reusing, and $|\mathcal{B}|$ is the buffer size.

## 3.2 ADAPTIVE TRAINING BATCH CONSTRUCTION

The core of off-policy RLVR lies in how to integrate historical experiences with online samples, to maintain non-homogeneous rewards and an appropriate difficulty distribution in each training step. For BAPO, we introduce **a filter function** $I(x)$ in Definition 3.1 that decomposes the data selection criteria for each training step's batch into three parts.

**Definition 3.1** (Training Batch Filtering Function). Define $\mu_{\pi,r}(x) = \mathbb{E}_{y\sim\pi(\cdot|x)}[r(x,y)]$ as the expected reward under policy $\pi$ for input $x$. The training batch indicator function $I : \mathcal{X} \to \{0,1\}$ is formulated as:

$$I(x) = \underbrace{\mathbb{1}_{\{\frac{1}{G}\leq\mu_{\alpha,r}(x)\leq\frac{G-1}{G}\}}}_{\text{Filtered Fresh}} + \underbrace{\mathbb{1}_{\{\mu_{\alpha_\mathcal{B},r}(x)\leq c_1 \wedge \mu_{\pi_{\theta_t},r}(x)>c_1\}}}_{\text{Improved Historical Difficult}} + \underbrace{\mathbb{1}_{\{c_2\leq\mu_{\alpha_\mathcal{B},r}(x)\leq c_3\}}}_{\text{Historical High-quality}} \quad (5)$$

where $\alpha$ denotes the delayed rollout policy and $\alpha_\mathcal{B}$ denotes the policy associated with buffer samples. The function selects samples based on three criteria, yielding subsets $\mathcal{X}_1$, $\mathcal{X}_2$ and $\mathcal{X}_3$ respectively.

Next, we explain the selection principles for $I(x)$ and derive three categories of samples, namely $\mathcal{X}_1$, $\mathcal{X}_2$, and $\mathcal{X}_3$, which are obtained from these three conditions, respectively.

**(1) Filtered Fresh Samples ($\mathcal{X}_1$).** To prevent gradient vanishing and maintain training stability, we filter the online rollout batch to exclude samples with zero variance. Specifically, we retain fresh

samples where the group mean reward satisfies $\mu_{\alpha,r}(x) \in [\frac{1}{G}, \frac{G-1}{G}]$. While other filtering strategies (e.g., Gaussian sampling or uniform sampling) can be applied, we find that simple truncation sufficient for effective learning. A detailed discussion and comparison of different online filtering functions are provided in Appendix A.3.

**(2) Improved Historical Difficult Samples ($\mathcal{X}_2$).** Samples exhibiting extremely low group mean rewards, where $\mu_{\alpha,r}(x) \in [0, c_1]$, present significant challenges to the current policy and typically yield negligible policy improvement. However, as the model evolves, these historically difficult queries may eventually become tractable for a successor policy. To harness this, we *periodically* re-generate responses using the current policy $\pi_{\theta_t}$ every $m$ training steps and construct the subset $\mathcal{X}_2$ based on the observable improvement.

Let $\mathcal{B}_{\text{bad}} \subseteq \mathcal{B}$ denote the buffer for difficult samples. To manage the computational overhead associated with the re-evaluation process, we limit the buffer capacity $|\mathcal{B}_{\text{bad}}|$ to be equal to the training batch size. A First-In-First-Out (FIFO) mechanism is employed to automatically discard outdated samples when the buffer reaches capacity. $\mathcal{X}_2$ is formulated as:

$$\mathcal{X}_2 = \left\{ (x, y') \mid (x, y) \in \mathcal{B}_{\text{bad}}, y' \sim \pi_{\theta_t}(\cdot \mid x), c_1 < \mu_{\pi_{\theta_t}, r}(x) < 1 \right\} \tag{6}$$

where $y'$ represents the new response generated by $\pi_{\theta_t}$, and we specifically select samples that show improvement such that $c_1 < \mu_{\pi_{\theta_t}, r}(x) < 1$.

**(3) Reused Historical High-quality Samples ($\mathcal{X}_3$).** To prevent underfilled batches caused by the scarcity of $\mathcal{X}_1$ and $\mathcal{X}_2$, we maintain a FIFO auxiliary buffer $\mathcal{B}_{\text{high}} \subseteq \mathcal{B}$. To mitigate training instability from stale data, $\mathcal{B}_{\text{high}}$ is restricted to high-quality trajectories from the three most recent steps. The subset $\mathcal{X}_3$ is randomly sampled to fill the remaining capacity:

$$\mathcal{X}3 = \mathcal{S}\left(\mathcal{B}_{\text{high}}, \min\left(|\mathcal{B}_{\text{high}}|, B - |\mathcal{X}_1| - |\mathcal{X}_2|\right)\right) \tag{7}$$

where $B$ is the configured training batch size and $\mathcal{S}(\cdot, k)$ denotes the random sampling of $k$ elements. Furthermore, to progressively master increasingly difficult tasks, we employ a linear mapping to shift the historical "high-quality" from easier to harder instances, scaling in accordance with the global average performance $r_{\text{tot}}$:

$$c_i = r_{\text{tot}} \cdot (c_i^{\text{high}} - c_i^{\text{low}}) + c_i^{\text{low}}, \quad i \in 2, 3 \tag{8}$$

### 3.3 THEORETICAL ANALYSIS

In this section, we further provide theoretical analysis in Theorem 3.2 to establish BAPO's training stability based on (Mroueh et al., 2025)'s theorem. We show that, under certain assumptions, our constructed adaptive batches can consistently maintain guaranteed bounded policy improvement.

**Theorem 3.2 (Policy Improvement Lower Bound with Adaptive Training Batch).** *Assume rewards are bounded: $0 \leq r \leq 1$. Let $\pi_{\theta_t}$ be the current policy, $\alpha_1 = \pi_{\theta_{t-v}}$ be the delayed rollout policy, $\alpha_2 = \pi_{\theta_t}$ be the current policy for re-evaluation, $\alpha_3 = \alpha_{\mathcal{B}}$ be the buffer policy distribution, and $I(x)$ be the filtering function partitioning samples into $\mathcal{X}_1$, $\mathcal{X}_2$, and $\mathcal{X}_3$.*

*Suppose $c_1, c_2, c_3 \in (0, 1)$ with $c_2 < c_3$, and the following TV distance constraints hold:*

$$TV(\pi_{\theta_t}(\cdot|x), \pi_{\theta_{t-v}}(\cdot|x)) \leq \delta_1 \quad \forall x \in \mathcal{X}_1 \tag{9}$$

$$TV(\pi_{\theta_t}(\cdot|x), \alpha_{\mathcal{B}}(\cdot|x)) \leq \delta_3 \quad \forall x \in \mathcal{X}_3 \tag{10}$$

*where $\delta_1, \delta_3 > 0$ are sufficiently small such that the variance lower bounds remain positive.*

*Then, for the policy update objective in Equation 3, the expected policy improvement over filtered samples satisfies:*

$$\mathbb{E}_{x \sim \rho_{\mathcal{X}}}[I(x)(J(\pi_\theta(\cdot|x)) - J(\pi_{\theta_t}(\cdot|x)))] \geq \sum_{i=1}^{3} \mathcal{L}_i(\pi_\theta, \alpha_i)$$

*where:*

$$J(\pi_\theta(\cdot \mid x)) = \mathbb{E}_{y \sim \pi_\theta(\cdot|x)} r(x, y)$$

$$\mathcal{L}_i(\pi_\theta, \alpha_i) = \mathbb{E}_{x \in \mathcal{X}_i}[L_{\alpha_i}(\pi_\theta(\cdot|x)) - 2K_i \cdot TV(\pi_\theta(\cdot|x), \alpha_i(\cdot|x)) - 2TV(\pi_{\theta_t}(\cdot|x), \alpha_i(\cdot|x))]$$

with $L_{\alpha_i}(\pi_\theta(\cdot|x)) = \frac{1}{\sigma_{\alpha_i,r,\varepsilon}(x)}(J(\pi_\theta(\cdot|x)) - J(\alpha_i(\cdot|x)))$. *The constants are:*

$$K_1 = \frac{1 - \sqrt{\frac{G-1}{G^2} + \varepsilon}}{\sqrt{\frac{G-1}{G^2} + \varepsilon}} \tag{11}$$

$$K_2 = \frac{1 - \sqrt{c_1(1 - c_1) + \varepsilon}}{\sqrt{c_1(1 - c_1) + \varepsilon}} \tag{12}$$

$$K_3 = \frac{1 - \sqrt{\min(c_2(1 - c_2), c_3(1 - c_3)) + \varepsilon}}{\sqrt{\min(c_2(1 - c_2), c_3(1 - c_3)) + \varepsilon}} \tag{13}$$

More importantly, we highlight several properties from this theorem:

**Bounded Stability.** All constants $K_1$, $K_2$, and $K_3$ are finite positive values, which guarantee that the training process remains numerically stable and theoretically bounded.

**Off-policy Tolerance.** The stability of trust-region methods inherently constrain the magnitude of single-step policy updates. Consequently, the divergence between the current policy $\pi_{\theta_t}$ and the delayed rollout policy $\alpha$ remains bounded over short intervals. Furthermore, the strict FIFO mechanism with limited buffer capacity ensures that only samples from recent policies are retained, thereby maintaining policy consistency within the training batch.

## 4 EXPERIMENTAL SETUP

To comprehensively evaluate the effectiveness of our off-policy RLVR framework, we conduct extensive experiments across different tasks and backbones, following the experimental setup described in (Qu et al., 2025).

First, we select three representative reasoning tasks, as detailed below:

**Mathematics.** Following prior work (Luo et al., 2025), we use the DeepSeek R1 Distilled 1.5B (Guo et al., 2025) and Qwen3 8B (Yang et al., 2025) as the base model, and conducted post-training on the DeepScaleR-Preview-Dataset (Aggarwal & Welleck, 2025), which contains 40 thousand question-answer pairs sourced from several mathematics competitions. Evaluation is performed on multiple mathematics benchmarks, including AIME24, AMC23, MATH500 (Hendrycks et al., 2021), Minerva Math (Minerva) (Lewkowycz et al., 2022), and OlympiadBench (Olympiad)He et al. (2024).

**Planning.** We choose Qwen2.5 Math 1.5B and 7B (Yang et al., 2024) as the backbone, and adopted the Countdown Number Game as the specific task. For training, we used a 10,000-problem subset of the Countdown-34 dataset, where each problem provides 3-4 source numbers. Evaluation was conducted on two variants: Countdown-3to4 (CD-34) test set using a 200-problem held-out split, and the more challenging Countdown-4 (CD-4) test set with 200 problems that consistently provide four source numbers (Chen et al., 2025b).

**Visual Geometry.** We train Qwen2.5 VL 3B and 7B (Bai et al., 2025) on the 2,101-problem training split of the Geometry3K dataset (Lu et al., 2021), where each problem consists of a geometric diagram paired with a natural language question requiring spatial and logical reasoning. Evaluation was performed on the official 300-problem validation split (Geo-3K val) and 601-problem test split of Geometry3K (Geo-3K test).

Besides, we select several on-policy and off-policy RLVR frameworks as baselines:

**On-policy.** We select GRPO (Shao et al., 2024), DAPO (Yu et al., 2025), and MoPPS (Qu et al., 2025) as representative on-policy RLVR methods. GRPO is the first to integrate group-relative advantage estimation into the RLVR framework, while DAPO further improves training stability and efficiency. MoPPS incorporates difficulty-aware prediction into prompt selection.

**Off-policy.** We compare our approach with three representative off-policy methods: GRPO ($v = 5$) (Mroueh et al., 2025), RePO (Li et al., 2025), and Remix-GRPO (Liang et al., 2025). Specifically, GRPO ($v = 5$) delays the rollout policy with a frequency of 5, whereas RePO and Remix-GRPO adopt diverse replay strategies to retrieve off-policy samples from a replay buffer.

**Implementation Details.** All comparative experiments were run on 8 A100 GPUs with 80GB memory based on the Verl framework (Sheng et al., 2025). Identical parameters were used to ensure fair comparison, with specific details in Appendix A.7.

# 5 RESULTS ANALYSIS

## 5.1 MAIN RESULTS

We evaluate BAPO across three reasoning tasks to demonstrate its broad applicability. Experimental results show that BAPO consistently outperforms existing baselines throughout training (Figure 4) and testing (Figure 12). Notably, in mathematical tasks, the GRPO baseline exhibits severe training instability, as evidenced by significant oscillations in its early-stage training curve. This is attributed to the high variance in problem difficulty within the DeepScalerR dataset. Under the same settings, BAPO achieves smoother convergence and higher reward bounds.

In Tables 1, BAPO achieves an **average 12.5% accuracy improvement** over baselines. Crucially, while DAPO approaches BAPO's performance in some metrics, it requires approximately **2.5×** **more rollouts** (as visualized in Figure 9), imposing a substantial computational burden.

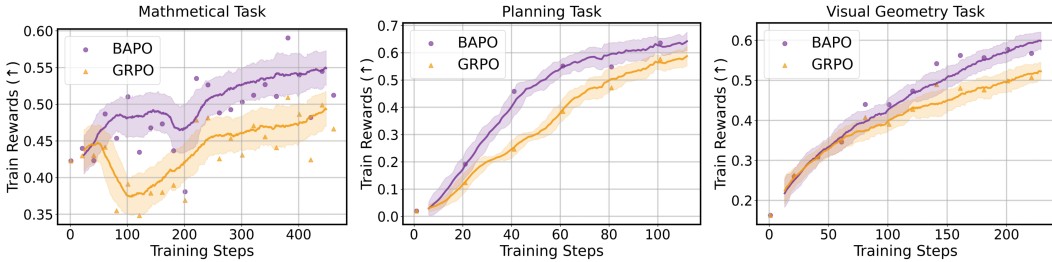

Figure 4: **Training Curves of Reward Changes** for mathematics, planning, and geometry tasks using DeepSeek Distilled Qwen 1.5B, Qwen2.5 Math 1.5B, and Qwen2.5 VL 3B, respectively.

Table 1: **Comprehensive Evaluation Results.** '+' indicates fine-tuning via the corresponding method. Accuracy is averaged over 32 runs. The **bold** value denotes the top result, and the underlined value denotes the second-top result.

| (a) Mathematics Benchmarks | | | | | | | | |
|---|---|---|---|---|---|---|---|---|
| **Method** | **AIME24** | **AMC** | **MATH500** | **Minerva.** | **Olympiad.** | **Avg. ↑** | **Rollouts ↓** | **Type** |
| **DeepSeek R1 Distill Qwen 1.5B** | 28.80 | 62.90 | 82.80 | 26.50 | 44.42 | 48.90 | - | - |
| **+GRPO (Guo et al., 2025)** | 30.73 | 67.47 | 85.40 | 28.95 | 45.33 | 51.58 | **677k** | on |
| **+DAPO (Yu et al., 2025)** | 35.73 | 70.08 | 86.05 | **30.70** | 48.48 | 54.20 | 1921k | on |
| **+MoPPS* (Qu et al., 2025)** | 33.33 | 65.29 | 84.94 | 28.88 | 45.93 | 51.67 | 737k | on |
| **+GRPO ($v=5$) (Mroueh et al., 2025)** | 30.49 | 65.09 | 86.72 | 28.16 | 46.18 | 51.57 | **677k** | off |
| **+RePO (Li et al., 2025)** | 30.42 | 64.76 | 83.75 | 28.33 | 45.44 | 50.54 | **677k** | off |
| **+Remix-GRPO* (Liang et al., 2025)** | 33.33 | 65.06 | 84.60 | 26.10 | 43.55 | 50.53 | - | off |
| **+BAPO (Ours)** | **38.54** | **72.74** | **89.18** | 29.55 | **50.06** | **56.01** | 733k | off |

| (b) Planning and Visual Geometry Benchmarks | | | | | | | |
|---|---|---|---|---|---|---|---|
| **Method** | **CD-34** | **CD-4** | **Avg** | **Method** | **Geo-3K(val)** | **Geo-3K(test)** | **Avg** |
| **Qwen2.5 Math 1.5B** | 1.12 | 0.37 | 0.75 | **Qwen2.5 VL 3B** | 14.77 | 19.18 | 16.98 |
| **+GRPO (Guo et al., 2025)** | 62.94 | 35.88 | 49.41 | **+GRPO (Guo et al., 2025)** | 36.44 | 43.12 | 39.78 |
| **+DAPO (Yu et al., 2025)** | 70.56 | 45.87 | 58.22 | **+DAPO (Yu et al., 2025)** | **40.11** | 45.18 | 42.65 |
| **+BAPO w/o $\mathcal{X}_2$ (Ours)** | 60.31 | 35.31 | 47.81 | **+BAPO w/o $\mathcal{X}_2$ (Ours)** | 30.57 | 36.92 | 33.75 |
| **+BAPO w/o $\mathcal{X}_3$ (Ours)** | 64.43 | 38.75 | 51.59 | **+BAPO w/o $\mathcal{X}_3$ (Ours)** | 32.22 | 39.79 | 36.01 |
| **+BAPO (Ours)** | **73.00** | **47.50** | **60.25** | **+BAPO (Ours)** | **40.11** | **46.33** | **43.22** |
| **Qwen2.5 Math 7B** | 2.68 | 0.94 | 1.81 | **Qwen2.5 VL 7B** | 30.40 | 36.10 | 33.25 |
| **+GRPO (Guo et al., 2025)** | 70.75 | 50.25 | 60.50 | **+GRPO (Guo et al., 2025)** | 40.79 | 47.15 | 43.97 |
| **+DAPO (Yu et al., 2025)** | 78.75 | **57.43** | 68.09 | **+DAPO (Yu et al., 2025)** | 40.87 | 47.02 | 43.95 |
| **+BAPO (Ours)** | **79.13** | 57.13 | **68.13** | **+BAPO (Ours)** | **41.89** | **48.77** | **45.33** |

*This method's performance is taken from the corresponding paper.

## 5.2 MECHANISM ANALYSIS

To deeply investigate whether BAPO's success stems from sensitive hyperparameter tuning or its core batch reconstruction mechanism, we conducted both **Minimalist Verification** and **Hyperparameter Robustness** experiments.

> **Off-policy Components > Off-policy Hyperparameters**
>
> The performance gains of BAPO primarily stem from the structural logic of its off-policy components rather than specific hyperparameter settings. The framework remains effective even under rigid, parameter-free conditions.

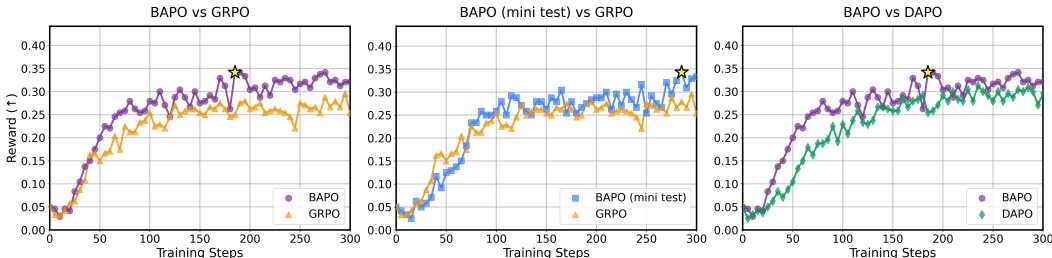

Figure 5: **Test Curves of Group Accuracy Changes** on AIME for different RLVR methods based on **Qwen3 8B**. Left: Standard BAPO vs. GRPO. Medium: **BAPO (mini test) vs. GRPO**. Right: Standard BAPO vs. DAPO.

**Minimalist Verification.** To validate the theoretical implications of Theorem 3.2 without relying on hyperparameter engineering, specifically avoiding the tuning of thresholds $c_1, c_2, c_3$ and update frequencies, we devised a **"Mini-test"** experiment. We trained Qwen3 8B on the mathematics task under 4K length constraints using a stripped-down, parameter-free BAPO logic for constructing training batch:

$\mathcal{X}_1$: We apply strictly standard zero-advantage filtering, removing only the prompts where all $G$ responses are entirely correct or entirely wrong.

$\mathcal{X}_2$: We replay historical *all-wrong* samples ($\mu_{\alpha,r}(x) = 0$). These correspond exactly to the difficult cases discarded by $\mathcal{X}_1$, creating a closed-loop system that recovers waste data without requiring a difficulty threshold $c_1$.

$\mathcal{X}_3$: Instead of a dynamic accuracy range, we reuse historical samples with exactly **50% accuracy**. As formally proven in Proposition A.3, samples with accuracy $\mu_{\alpha,r}(x) = \frac{1}{2}$ maximize the reward variance, thereby providing the theoretical maximum potential for single-step policy improvement $J(\pi_\theta) - J(\pi_{\theta_t})$.

The results in Figure 5 demonstrate that even in the hyperparameter-free "Mini-test", BAPO maintains a clear advantage over GRPO. This confirms that the structural introduction of $\mathcal{X}_2$ and $\mathcal{X}_3$ drives the performance, not the specific tuning of $c$ values.

**Component Efficacy.** To evaluate the contribution of re-evaluated difficult samples $\mathcal{X}_2$ and reused high-quality samples $\mathcal{X}_3$, we conduct ablation studies shown in Table 1 and Figure 6 (Column 2). Both components are essential: removing $\mathcal{X}_2$ causes a $\sim 21\%$ performance drop, underscoring the importance of explicitly targeting difficult samples.

**Hyperparameter Robustness.** We further evaluate the sensitivity of BAPO to its key hyperparameters: rollout delay $v$, re-rollout frequency $m$, and difficulty thresholds. **Frequency ($v, m$):** As shown in Figure 6 (Column 1), performance remains stable within reasonable ranges (e.g., $v = 5, m = 5$). Extreme delays only degrade performance when policy divergence becomes excessive, aligning with our theoretical analysis regarding the trust region. **Difficulty Thresholds ($c_2, c_3$):** While our adaptive boundary mechanism yields the best convergence, Figure 6 (Column 3) shows that using fixed

ranges still significantly outperforms baselines. This indicates that the *presence* of diverse historical data is more critical than the precise values of the thresholds.

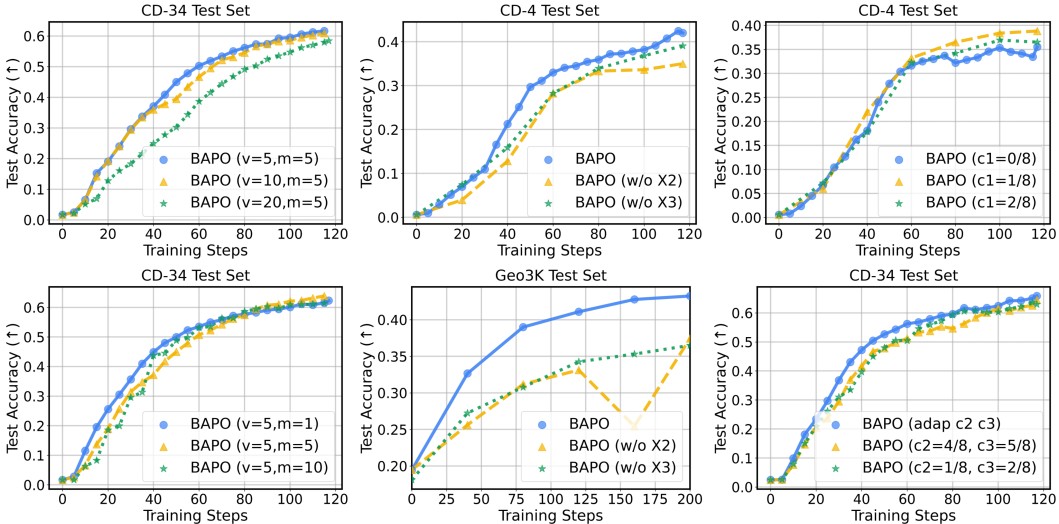

Figure 6: **Ablation Studies** for BAPO. The first column presents ablations on frequency-related hyperparameters $(m, v)$. The second column shows ablations on buffer subsets $(\mathcal{X}_2, \mathcal{X}_3)$. The third column compares fixed vs. adaptive difficulty thresholds.

## 5.3 DETAILED ANALYSIS

We analyze BAPO's internal mechanisms below. For extended analysis on training dynamics, computation, and visualization, please refer to **Appendices A.4, A.5 and A.6**.

**Tracking Difficult Samples.** We visualize the training dynamics in Figure 7. BAPO exhibits a superior capability to "unlock" difficult problems: after 3 epochs, BAPO successfully improves **31%** of the samples that were initially unsolvable ($0/8$ accuracy), compared to only **19%** for GRPO.

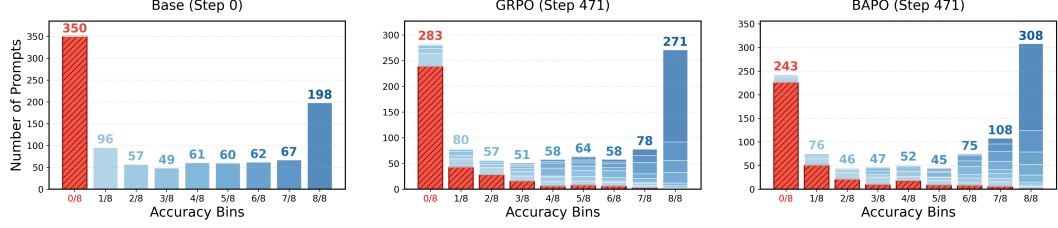

Figure 7: Tracking changes in the **Number of Different Accuracy Bins** on the DeepScalerR training subset. Special attention is paid to the reduction of bad samples (red bars).

**Sample Distribution & Efficiency.** To uncover the source of BAPO's efficiency, we analyze the dynamic batch construction in Figure 8 alongside the rollout costs in Figure 9.

As observed in Figure 8, the assembled training batch size frequently fluctuates below the maximum configured capacity. This reduction in backward propagation load effectively offsets the computational overhead caused by off-policy re-evaluation and log-probability re-computation. Consequently, as detailed in Table 2, BAPO maintains a training speed comparable to GRPO while requiring significantly fewer rollouts than DAPO, achieving a superior trade-off between convergence performance and computational cost.

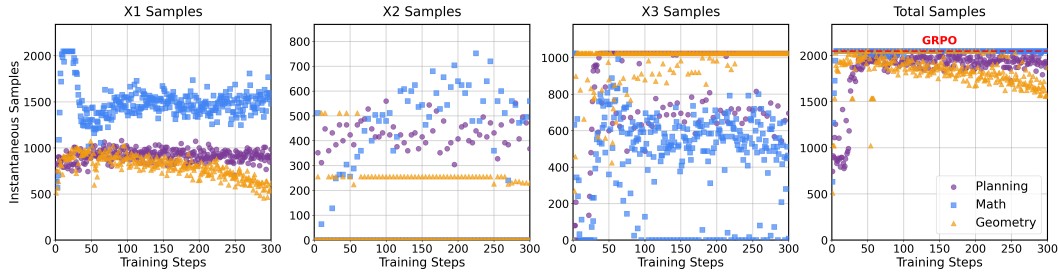

Figure 8: **Dynamic Sample Distribution.** The composition of BAPO's $\mathcal{X}_1, \mathcal{X}_2, \mathcal{X}_3$ and the total samples compared to the fixed GRPO batch size (Red line).

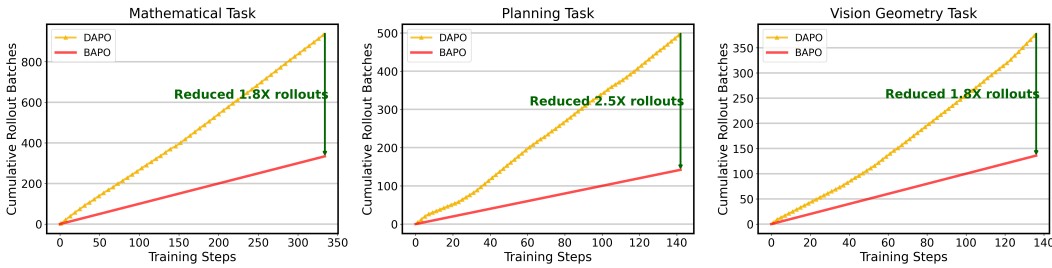

Figure 9: **Cumulative Rollout Batches Comparison between BAPO and DAPO.** The maximum rollout time for DAPO is set to 4.

---

**Efficient Batch Adaptation**

BAPO maintains training efficiency comparable to GRPO. While the periodic re-evaluation of $\mathcal{X}_2$ introduces additional generation overhead, this cost is effectively offset by reduced training samples, particularly during the initial stages of training.

---

## 6 CONCLUSION

In this paper, we propose BAPO, an off-policy RLVR framework for LLM post-training. It aims to utilize historical training data better and thereby improve training efficiency. Specifically, we appropriately delay the rollout policy to stabilize the policy discrepancies of buffer samples. More importantly, we construct training batches by re-evaluating difficult samples and reusing historical high-quality ones, thereby enhancing the efficiency of post-training. We validate the strong adaptability of the BAPO framework through experiments on three distinct reasoning tasks using different LLM backbones, and the results demonstrate that BAPO significantly outperforms baselines in both convergence performance and training efficiency. Nevertheless, exploring how to adapt BAPO to large models with MoE architectures, as well as to agentic RL frameworks, remains a significant challenge.

ACKNOWLEDGEMENTS

This work was supported by the National Natural Science Foundation of China under Grant 72571007 and Grant 72595830/72595831, and by Beijing Nova Program (No. 20250484850).

ETHICS STATEMENT

All authors of this study strictly adhere to the ICLR code of ethics. Our research does not involve any potential conflicts of interest or sponsorship issues. We have carefully considered and addressed concerns related to discrimination, bias, and fairness in our methodology. The study raises no privacy or security concerns, maintains full legal compliance, and upholds the highest standards of research integrity. All experimental procedures and data handling practices follow established ethical guidelines for machine learning research.

REPRODUCIBILITY STATEMENT

To ensure full reproducibility of our results, we provide comprehensive implementation details of the proposed BAPO training algorithm in the supplementary materials. All experimental settings, hyperparameters, and dataset specifications are clearly documented. For our theoretical contributions, complete proofs and clear explanations of all assumptions are included in the appendix. Code and data will be made available upon acceptance to facilitate replication of our findings.

THE USE OF LARGE LANGUAGE MODELS

In this research, we employed LLMs solely as language editing tools to improve the clarity and readability of our manuscript. LLMs were used for grammar checking, style refinement, and language polishing purposes only. All core research ideas, experimental design, analysis, and conclusions are entirely the original work of the authors. The use of LLMs did not contribute to the conceptual or technical content of this study.

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

# A APPENDIX

## A.1 GLOSSARY OF TERMS AND NOTATIONS

| Term | Definition |
| --- | --- |
| $c_1, c_2, c_3$ | Thresholds for classifying historical samples by difficulty (group mean reward). |
| $\mathcal{X}_1, \mathcal{X}_2, \mathcal{X}_3$ | Subsets of training batch: fresh, re-evaluated difficult, and historical high-quality samples. |
| $m$ | Re-evaluation frequency for historically difficult samples. |
| $v$ | Delay steps for updating the rollout policy. |
| $G$ | Group size, number of responses generated per prompt during rollout. |
| $\mathcal{B}$ | Replay buffer storing historical samples. |
| $\hat{A}_{i,t}$ | Estimated advantage for token $t$ in response $i$. |
| $\varepsilon$ | Clipping parameter in PPO-style objectives. |
| $\beta$ | Coefficient for KL penalty in the objective function. |
| $I(x)$ | Filter function for constructing BAPO's training batch. |
| $\mathbb{D}_{\mathrm{KL}}$ | Kullback–Leibler divergence, used to constrain policy deviation. |
| $\alpha$ | Rollout policy for BAPO, which synchronizes to $\pi_\theta$ every $v$ steps. |
| $\pi_\theta$ | LLM policy parameterized by $\theta$. |
| $\pi_{\mathrm{ref}}$ | Reference policy (e.g., initial pre-trained model). |
| $\rho(\theta)$ | Importance sampling ratio: $\frac{\pi_\theta(y|x)}{\pi_{\mathrm{old}}(y|x)}$. |
| $r(x, y)$ | Reward function, we set to binary (0/1) based on correctness. |
| $\mu_{\alpha,r}(x)$ | Expected reward under policy $\pi$ for input $x$. We approximate this value using the mean of $r(x, y)$ corresponding to $G$ responses $y$ generated by the rollout policy $\alpha$ for each prompt $x$. |
| $\sigma_{\alpha,r,\varepsilon}(x)$ | Standard deviation of rewards under policy $\alpha$ for input $x$, with smoothing $\varepsilon$. |
| $J(\pi(\cdot|x))$ | Expected reward of policy $\pi$ for input $x$: $\mathbb{E}_{y\sim\pi(\cdot|x)}[r(x,y)]$. |
| $\mathcal{N}(\mu_{\alpha,r}(x) \mid \mu, \sigma^2)$ | A sampling method that assigns weights to online rollouts based on a normal distribution centered at $\mu$ with standard deviation $\sigma$, used to filter samples by their group mean reward $\mu_{\alpha,r}(x)$. |

## A.2 THEORETICAL ANALYSIS

**Lemma A.1** (Kantorovich-Rubenstein duality of total variation distance)**.** *The Kantorovich-Rubinstein duality (variational representation) of the total variation distance is as follows:*

$$TV(m_1, m_2) = \frac{1}{2L} \sup_{g \in \mathcal{G}_L} \left\{ \mathbb{E}_{Z \sim m_1}[g(Z)] - \mathbb{E}_{Z \sim m_2}[g(Z)] \right\}, \tag{14}$$

*where $\mathcal{G}_L = \{g : \mathcal{Z} \to \mathbb{R}, \|g\|_\infty \le L\}$.*

**Theorem A.2** (**Policy Improvement Lower Bound with Adaptive Training Batch**)**.** *Assume rewards are bounded: $0 \le r \le 1$. Let $\pi_{\theta_t}$ be the current policy, $\alpha_1 = \pi_{\theta_{t-v}}$ be the delayed rollout policy, $\alpha_2 = \pi_{\theta_t}$ be the current policy for re-evaluation, $\alpha_3 = \alpha_{\mathcal{B}}$ be the buffer policy distribution, and $I(x)$ be the filtering function partitioning samples into $\mathcal{X}_1$, $\mathcal{X}_2$, and $\mathcal{X}_3$.*

*Suppose $c_1, c_2, c_3 \in (0, 1)$ with $c_2 < c_3$. and the following TV distance constraints hold:*

$$TV(\pi_{\theta_t}(\cdot|x), \pi_{\theta_{t-v}}(\cdot|x)) \le \delta_1 \quad \forall x \in \mathcal{X}_1 \tag{15}$$

$$TV(\pi_{\theta_t}(\cdot|x), \alpha_{\mathcal{B}}(\cdot|x)) \le \delta_3 \quad \forall x \in \mathcal{X}_3 \tag{16}$$

*where $\delta_1, \delta_3 > 0$ are sufficiently small such that the variance lower bounds remain positive.*

*Then, for the policy update objective in Equation 3, the expected policy improvement over filtered samples satisfies:*

$$\mathbb{E}_{x \sim \rho_{\mathcal{X}}}[I(x)(J(\pi_\theta(\cdot|x)) - J(\pi_{\theta_t}(\cdot|x)))] \ge \sum_{i=1}^3 \mathcal{L}_i(\pi_\theta, \alpha_i)$$

*where:*

$$\mathcal{L}_i(\pi_\theta, \alpha_i) = \mathbb{E}_{x \in \mathcal{X}_i} \left[ L_{\alpha_i}(\pi_\theta(\cdot|x)) - 2K_i \cdot TV(\pi_\theta(\cdot|x), \alpha_i(\cdot|x)) - 2TV(\pi_{\theta_t}(\cdot|x), \alpha_i(\cdot|x)) \right]$$

*with* $L_{\alpha_i}(\pi_\theta(\cdot|x)) = \frac{1}{\sigma_{\alpha_i, r, \varepsilon}(x)}(J(\pi_\theta(\cdot|x)) - J(\alpha_i(\cdot|x)))$. *The constants are:*

$$K_1 = \frac{1 - \sqrt{\frac{G-1}{G^2} + \varepsilon}}{\sqrt{\frac{G-1}{G^2} + \varepsilon}} \tag{17}$$

$$K_2 = \frac{1 - \sqrt{c_1(1 - c_1) + \varepsilon}}{\sqrt{c_1(1 - c_1) + \varepsilon}} \tag{18}$$

$$K_3 = \frac{1 - \sqrt{\min(c_2(1 - c_2), c_3(1 - c_3)) + \varepsilon}}{\sqrt{\min(c_2(1 - c_2), c_3(1 - c_3)) + \varepsilon}} \tag{19}$$

*Proof.* We prove the bound by analyzing each filtered sample set separately, applying off-policy policy improvement bounds tailored to the reference distribution used in each region.

**Step 1: Core inequality for off-policy samples.** For any $x$ such that $I(x) = 1$, we establish the fundamental inequality:

$$J(\pi_\theta(\cdot|x)) - J(\pi_{\theta_t}(\cdot|x)) \geq L_{\alpha_i}(\pi_\theta(\cdot|x)) - 2K_i \cdot \text{TV}(\pi_\theta(\cdot|x), \alpha_i(\cdot|x)) \tag{20}$$
$$- 2\text{TV}(\pi_{\theta_t}(\cdot|x), \alpha_i(\cdot|x)) \tag{21}$$

where $K_i = \frac{1 - \sigma_{\alpha_i, r, \varepsilon}(x)}{\sigma_{\alpha_i, r, \varepsilon}(x)}$ is a constant that depends on the variance of rewards in each filtered subset.

First, we expand the advantage objective. By definition:

$$L_{\alpha_i}(\pi_\theta(\cdot|x)) = \mathbb{E}_{y \sim \alpha_i(\cdot|x)} \left[ \frac{\pi_\theta(y|x)}{\alpha_i(y|x)} A_{\alpha_i}(x, y) \right] \tag{22}$$

$$= \mathbb{E}_{y \sim \alpha_i(\cdot|x)} \left[ \frac{\pi_\theta(y|x)}{\alpha_i(y|x)} \cdot \frac{r(x, y) - \mu_{\alpha_i, r}(x)}{\sigma_{\alpha_i, r, \varepsilon}(x)} \right] \tag{23}$$

$$= \frac{1}{\sigma_{\alpha_i, r, \varepsilon}(x)}(J(\pi_\theta(\cdot|x)) - J(\alpha_i(\cdot|x))) \tag{24}$$

Next, we establish the key algebraic identity relating $L_{\alpha_i}(\pi_\theta(\cdot|x))$ to $J(\pi_\theta(\cdot|x)) - J(\pi_{\theta_t}(\cdot|x))$:

$$L_{\alpha_i}(\pi_\theta(\cdot|x)) - (J(\pi_\theta(\cdot|x)) - J(\pi_{\theta_t}(\cdot|x))) \tag{25}$$

$$= \frac{1 - \sigma_{\alpha_i, r, \varepsilon}(x)}{\sigma_{\alpha_i, r, \varepsilon}(x)}(J(\pi_\theta(\cdot|x)) - J(\alpha_i(\cdot|x))) + (J(\pi_{\theta_t}(\cdot|x)) - J(\alpha_i(\cdot|x))) \tag{26}$$

**Application of Kantorovich-Rubenstein duality:** For bounded rewards with $\|r\|_\infty = 1$, the Kantorovich-Rubenstein duality Lemma A.1 provides:

$$|J(\pi_\theta(\cdot|x)) - J(\alpha_i(\cdot|x))| \leq 2 \cdot \text{TV}(\pi_\theta(\cdot|x), \alpha_i(\cdot|x)) \tag{27}$$
$$|J(\pi_{\theta_t}(\cdot|x)) - J(\alpha_i(\cdot|x))| \leq 2 \cdot \text{TV}(\pi_{\theta_t}(\cdot|x), \alpha_i(\cdot|x)) \tag{28}$$

Since $0 \leq r \leq 1$, we have $\sigma_{\alpha_i, r, \varepsilon}(x) < 1$, ensuring $K_i = \frac{1 - \sigma_{\alpha_i, r, \varepsilon}(x)}{\sigma_{\alpha_i, r, \varepsilon}(x)} \geq 0$. Combining these bounds yields the desired inequality.

**Step 2: Analysis for $\mathcal{X}_1$ (Filtered fresh samples).** For $x \in \mathcal{X}_1$, samples are generated by the delayed rollout policy $\alpha_1 = \pi_{\theta_{t-v}}$ and selected via Gaussian sampling with group-level accuracy $\mu_{\alpha_1, r}(x) \in \{\frac{1}{G}, \frac{2}{G}, \ldots, \frac{G-1}{G}\}$, excluding extremes $\{0, 1\}$.

**Variance analysis on discrete set:** For the variance function $f(p) = p(1 - p)$ over the discrete set $\{\frac{1}{G}, \frac{2}{G}, \ldots, \frac{G-1}{G}\}$, the minimum value occurs at the boundary points $p = \frac{1}{G}$ or $p = \frac{G-1}{G}$, both yielding $f(p) = \frac{G-1}{G^2}$. Therefore:

$$\sigma^2_{\alpha_1, r}(x) = \mu_{\alpha_1, r}(x)(1 - \mu_{\alpha_1, r}(x)) \geq \frac{G-1}{G^2} \tag{29}$$

Thus: $\sigma_{\alpha_1, r, \varepsilon}(x) \geq \sqrt{\frac{G-1}{G^2} + \varepsilon}$, yielding:

$$K_1 = \frac{1 - \sqrt{\frac{G-1}{G^2} + \varepsilon}}{\sqrt{\frac{G-1}{G^2} + \varepsilon}}$$

**Step 3: Analysis for $\mathcal{X}_2$ (Re-evaluated difficult samples).** For $x \in \mathcal{X}_2$, samples are generated by the current policy $\alpha_2 = \pi_{\theta_t}$ through re-evaluation of historically difficult samples. The selection criterion ensures that historically difficult samples ($\mu_{\alpha_B, r}(x) \leq c_1$) now achieve improved performance ($c_1 < \mu_{\pi_{\theta_t}, r}(x) < 1$) under the current policy.

Since these samples are directly generated by $\pi_{\theta_t}$, we have $\alpha_2 = \pi_{\theta_t}$, and the constraint $c_1 < \mu_{\pi_{\theta_t}, r}(x) < 1$ provides a natural lower bound, yielding:

$$\sigma_{\alpha_2, r}^2(x) = \mu_{\alpha_2, r}(x)(1 - \mu_{\alpha_2, r}(x)) > c_1(1 - c_1) \tag{30}$$

Therefore: $\sigma_{\alpha_2, r, \varepsilon}(x) > \sqrt{c_1(1 - c_1) + \varepsilon}$, giving us:

$$K_2 = \frac{1 - \sqrt{c_1(1 - c_1) + \varepsilon}}{\sqrt{c_1(1 - c_1) + \varepsilon}}$$

**Step 4: Analysis for $\mathcal{X}_3$ (Historical high-quality samples).** For $x \in \mathcal{X}_3$, samples are generated by historical buffer policies $\alpha_3 = \alpha_B$ with $\mu_{\alpha_B, r}(x) \in [c_2, c_3]$.

Since $\mu_{\alpha_3, r}(x)(1 - \mu_{\alpha_3, r}(x))$ achieves its minimum at the endpoints of the interval $[c_2, c_3]$:

$$\sigma_{\alpha_3, r}^2(x) \geq \min(c_2(1 - c_2), c_3(1 - c_3)) \tag{31}$$

Therefore: $\sigma_{\alpha_3, r, \varepsilon}(x) \geq \sqrt{\min(c_2(1 - c_2), c_3(1 - c_3)) + \varepsilon}$, yielding:

$$K_3 = \frac{1 - \sqrt{\min(c_2(1 - c_2), c_3(1 - c_3)) + \varepsilon}}{\sqrt{\min(c_2(1 - c_2), c_3(1 - c_3)) + \varepsilon}}$$

**Step 5: Combining the results.** Taking expectations over $x \sim \rho_{\mathcal{X}}$ and applying the indicator function decomposition:

$$\mathbb{E}_{x \sim \rho_{\mathcal{X}}}[I(x)(J(\pi_\theta(\cdot|x)) - J(\pi_{\theta_t}(\cdot|x)))] \tag{32}$$

$$= \sum_{i=1}^{3} \mathbb{E}_{x \sim \rho_{\mathcal{X}}}[\mathbf{1}_{\{x \in \mathcal{X}_i\}}(J(\pi_\theta(\cdot|x)) - J(\pi_{\theta_t}(\cdot|x)))] \tag{33}$$

$$\geq \sum_{i=1}^{3} \mathbb{E}_{x \in \mathcal{X}_i}[L_{\alpha_i}(\pi_\theta(\cdot|x)) - 2K_i \cdot \text{TV}(\pi_\theta(\cdot|x), \alpha_i(\cdot|x)) - 2\text{TV}(\pi_{\theta_t}(\cdot|x), \alpha_i(\cdot|x))] \tag{34}$$

$$= \sum_{i=1}^{3} \mathcal{L}_i(\pi_\theta, \alpha_i) \tag{35}$$

All constants $K_1, K_2, K_3$ are finite, since denominators are strictly positive by construction and numerators are bounded by 1 under $c_1, c_2, c_3 \in (0, 1)$, completing the proof. $\square$

**Proposition A.3.** *For binary reward tasks where $r(x, y) \in \{0, 1\}$, the contribution to the policy improvement lower bound is maximized when the expected group reward of the sample is $\mu = 0.5$.*

*Proof.* Recalling Theorem 3.2, the lower bound for policy improvement on a specific data distribution involves the constant $K$, which scales the penalty for policy divergence. The tightness of this bound is governed by the standard deviation of the rewards $\sigma_{\alpha, r}(x)$.

Due to advantage standardization $\hat{A} \propto \frac{1}{\sigma}$, the effective step size in the advantage estimation and consequently the gradient magnitude is proportional to the inverse of the standard deviation. However, in the context of the lower bound analysis in Theorem 3.2, the stability constant $K$ is defined as:

$$K(\mu) = \frac{1 - \sigma(\mu)}{\sigma(\mu)} \tag{36}$$

where a smaller $K$ indicates a tighter bound and thus a larger guaranteed improvement step. For a binary reward function $r \in \{0, 1\}$, the reward distribution follows a Bernoulli distribution with parameter $\mu(x) = \mathbb{E}[r|x]$. The variance is given by:

$$\sigma^2(\mu) = \mu(1 - \mu) \tag{37}$$

To find the $\mu$ that maximizes variance, we take the derivative with respect to $\mu$:

$$\frac{d}{d\mu}(\mu - \mu^2) = 1 - 2\mu \tag{38}$$

Setting the derivative to zero:

$$1 - 2\mu = 0 \implies \mu = 0.5 \tag{39}$$

Since the second derivative $\frac{d^2}{d\mu^2} = -2 < 0$, this is a global maximum.

At $\mu = 0.5$, the variance is maximized ($\sigma^2 = 0.25, \sigma = 0.5$). This corresponds to the state of maximum entropy, where the model is most "uncertain" about the outcome. Training on these samples provides the strongest gradient signal for distinguishing between correct and incorrect reasoning paths, effectively maximizing the information gain per step. Conversely, as $\mu \to 0$ or $\mu \to 1$, $\sigma \to 0$, causing the advantage estimates to numerical instability or the gradient signal to vanish. Therefore, selecting samples with $\mu = 0.5$ theoretically offers the most efficient learning signal and the most favorable stability bound.

$\square$

### A.3 ONLINE FILTER MECHANISM ANALYSIS

To investigate the impact of fresh sample selection on training stability and convergence, we conduct an ablation study using Qwen3 8B with a 4K response length limit. We compare three distinct filtering strategies for the online component ($\mathcal{X}_1$):

**Mode 1 (Range Filter):** It retains samples with group mean rewards $\mu \in [\frac{1}{G}, \frac{G-1}{G}]$. This effectively removes only the zero-advantage samples (all-correct or all-incorrect) that contribute minimal gradients.

**Mode 2 (Gaussian Filter):** A difficulty-weighted strategy that prioritizes samples with high variance (accuracy near 0.5) using a Gaussian distribution, thereby reducing the proportion of extremely easy or hard samples.

**Mode 3 (Uniform Filter):** A baseline that randomly selects 60% of the fresh samples regardless of their quality. This ratio was chosen to match the approximate data retention rates of Mode 1 and Mode 2 (approximately 40%–60%) for a fair comparison of data volume.

**The Value of Quality over Randomness.** As illustrated in Figure 10, the uniform filter mechanism exhibits severe instability, characterized by exploding gradient norms and a complete collapse in performance after 150 steps. Since this strategy blindly includes all-wrong samples (where $\mu = 0$), the model is forced to update based on low-quality, zero-advantage signals. Suppressing the token probabilities of incorrect responses without a corresponding positive signal introduces significant noise and uncertainty, ultimately destabilizing the policy. This failure highlights that the *quality* of the training batch, particularly the exclusion of zero-advantage noise, is crucial.

**Convergence Speed and Final Performance.** The Gaussian filter demonstrates faster convergence in the early stages. By focusing heavily on samples with the highest variance (accuracy $\approx 0.5$), it provides the steepest learning signal initially. However, its final convergence performance is lower

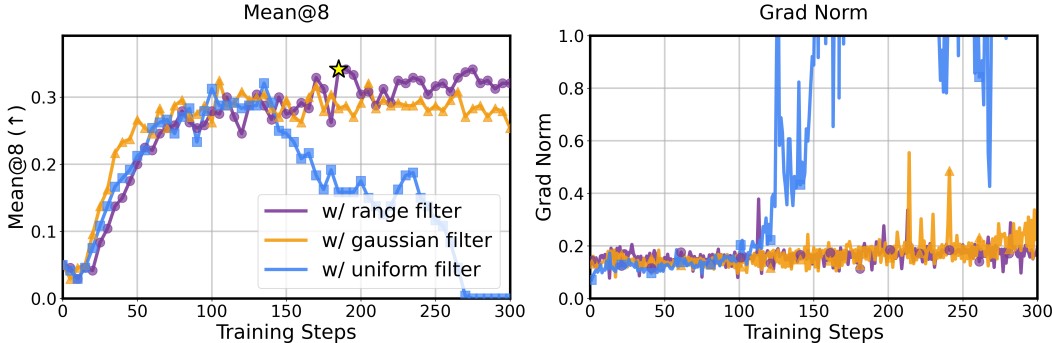

Figure 10: **Ablation on Online Filtering Strategies.** Comparison of Range Filter, Gaussian Filter, and Uniform Filter on training stability (Grad Norm) and performance (Mean@8). The star symbol indicates the best checkpoint for BAPO.

than that of the range filter. We hypothesize that the Gaussian filter restricts sample diversity by aggressively filtering out samples that are slightly easier or harder but still informative. In contrast, the range filter retains a broader spectrum of valid samples. While it learns slightly slower initially, it maintains a rich distribution of training data, preventing premature plateauing and ultimately achieving the highest asymptotic performance.

## A.4 TRAINING DYNAMICS AND TEST CURVES

As illustrated in Figure 11 and Figure 12, we present more detailed training dynamics and test curves for the Planning and Vision Geometry tasks. The results indicate that both BAPO and DAPO consistently outperform GRPO in terms of training rewards. Interestingly, BAPO exhibits higher entropy, reflecting better exploration capability compared to other algorithms, which also results in longer response lengths.

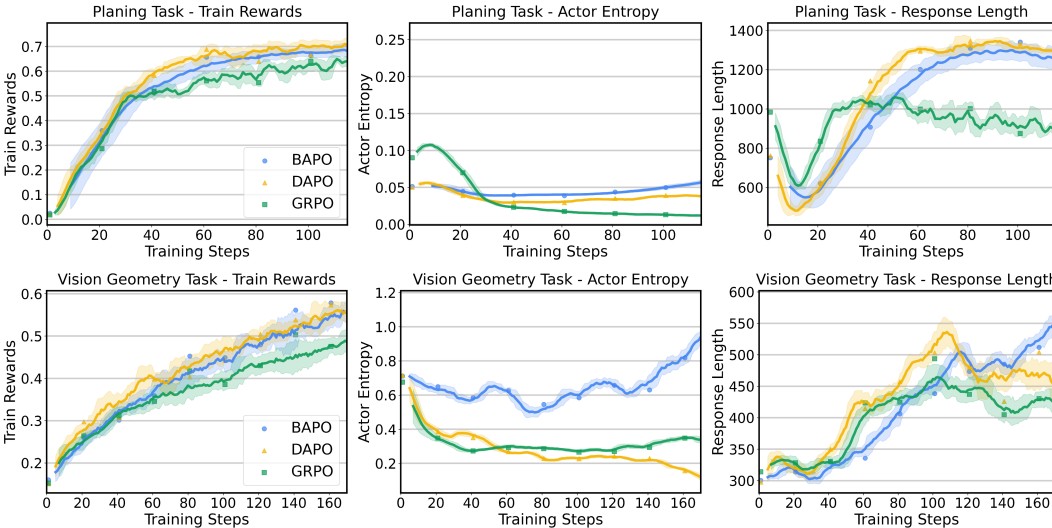

Figure 11: **Training Dynamics** during BAPO, GRPO, and DAPO post-training, including training rewards, training entropy, and response lengths.

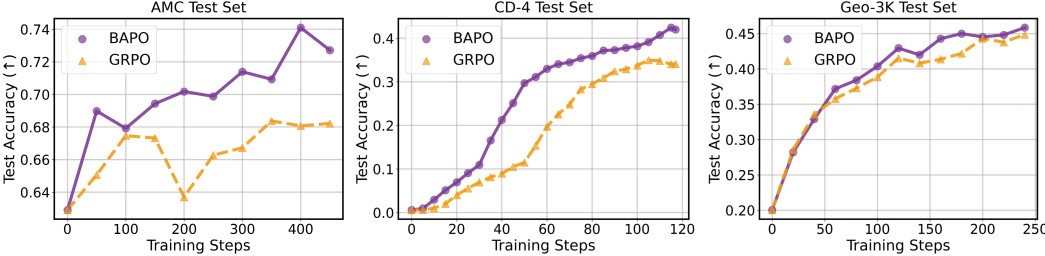

Figure 12: **Test Curves of Group Accuracy Changes** for mathematics, planning, and geometry tasks among AMC, CD-4 test set, and Geo-3K test set, respectively.

### A.5 COMPUTATION ANALYSIS

From Table 2, we observe that BAPO's computational overhead correlates with the number of samples requiring re-evaluation and the actual training batch size. For the Planning task, BAPO (w/o $\mathcal{X}_2$) achieves the fastest training time by eliminating bad case re-evaluation, but this comes at the cost of reduced performance. For the Mathematics task, the high number of bad cases (as shown by the 0/8 accuracy samples in Figure 7) means that under our re-evaluation frequency setting of $m = 5$, inference time exceeds that of GRPO. However, this additional time investment proves valuable, yielding better bad-case handling rates and overall test performance, as shown in Figure 4 and Table **??**. We plan to explore lower re-evaluation frequencies to assess the performance trade-offs.

BAPO ($c_2 = 0.375, c_3 = 0.5$) runs significantly faster than BAPO ($c_2 = 0, c_3 = 0.25$) due to the larger historical data volume in the latter configuration. This causes BAPO ($c_2 = 0, c_3 = 0.25$) to maintain a larger effective batch size than BAPO ($c_2 = 0.375, c_3 = 0.5$). Training logs also confirm this observation: BAPO ($c_2 = 0, c_3 = 0.25$) consistently utilizes 100% of the configured batch size (equivalent to on-policy methods' batch size), while BAPO ($c_2 = 0.375, c_3 = 0.5$) operates at approximately 70% capacity.

Table 2: **Computational Overhead Analysis.** "**Batch size**" $(a, b)$ represents the sample batch size $a$ and train mini batch size $b$. "**Time**" is measured in total training time (d=days, h=hours, m=minutes) on 8 A100 GPUs.

| Tasks | Methods | Batch Size | Num Epoch | Time |
|---|---|---|---|---|
| Mathematics | GRPO | (256, 64) | 3 | **1d 16h 58m** |
| | DAPO | (256, 64) | 3 | 2d 15h 30m |
| | **BAPO** | (256, 64) | 3 | 1d 22h 37m |
| Planning | GRPO | (256, 64) | 3 | 3h 47m |
| | DAPO | (256, 64) | 3 | 6h 35m |
| | **BAPO** | (256, 64) | 3 | 3h 23m |
| | **BAPO (w/o $\mathcal{X}_2$)** | (256, 64) | 3 | **2h 38m** |
| | **BAPO (w/o $\mathcal{X}_3$)** | (256, 64) | 3 | 3h 4m |
| | **BAPO ($c_2 = 0, c_3 = 0.25$)** | (256, 64) | 3 | 3h 54m |
| | **BAPO ($c_2 = 0.375, c_3 = 0.5$)** | (256, 64) | 3 | 3h 4m |
| Visual Geometry | GRPO | (256, 64) | 30 | 7h 55m |
| | DAPO | (256, 64) | 30 | 12h 19m |
| | **BAPO** | (256, 64) | 30 | 5h 50m |
| | **BAPO (w/o $\mathcal{X}_2$)** | (256, 64) | 30 | **3h 42m** |
| | **BAPO (w/o $\mathcal{X}_3$)** | (256, 64) | 30 | 4h 31m |

### A.6 VISUALIZATION

We present additional visualization details, including the sample accuracy tracking for the Countdown and Geometry3K datasets, as shown in Figure 13. Meanwhile, we visualize the source of

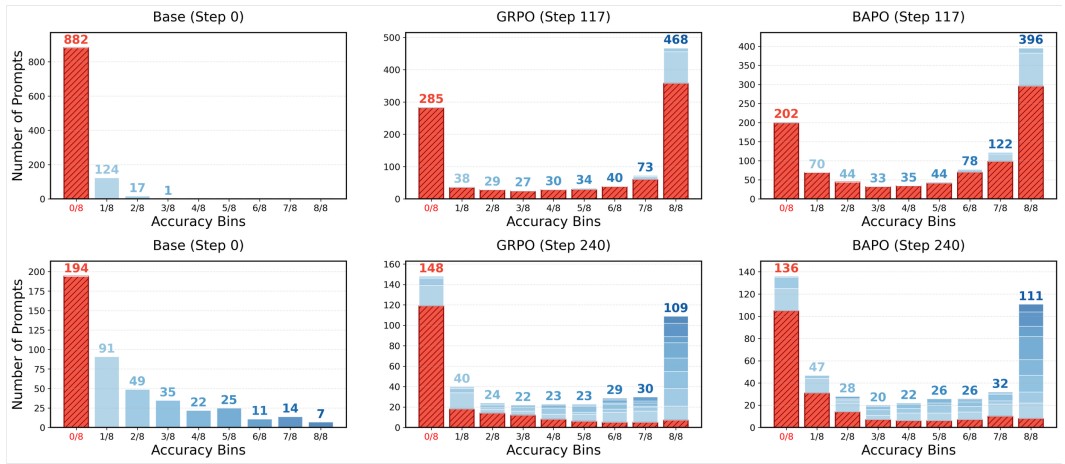

Figure 13: Tracking changes in the **Number of Different Accuracy Bins** on the Countdown (upper) and Geometry3K training sets (lower) for the baseline model, GRPO, and our BAPO method. Special attention is paid to the change in the number of bad samples (red bars) that the base model fails to handle.

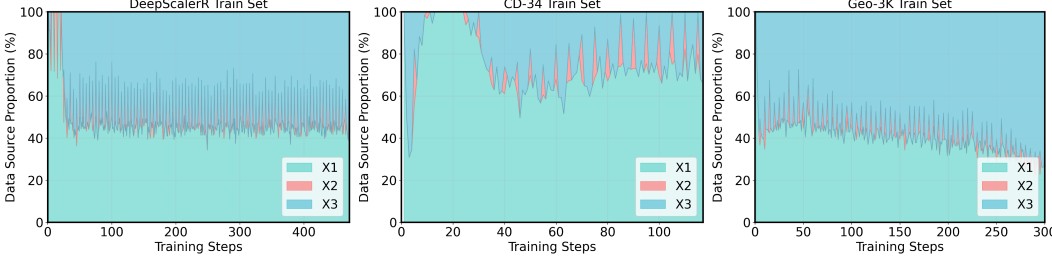

Figure 14: **Batch Distribution Visualization of** $\mathcal{X}_1$, $\mathcal{X}_2$, $\mathcal{X}_3$ for Mathematics, Planning, and Visual Geometry Tasks (left to right) during BAPO's training.

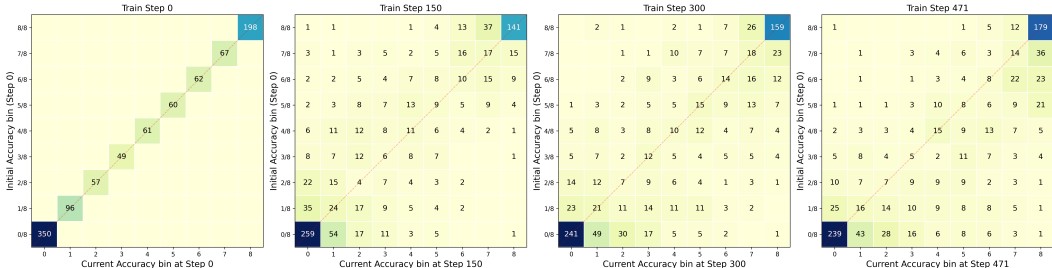

Figure 15: **Accuracy Migration Matrix Analysis.** We track a fixed subset of 1,000 randomly selected prompts from the training set and visualize their movement between accuracy bins (0/8 to 8/8) at Steps 0, 150, 300, and 471 (the last step). The y-axis represents the initial accuracy bin at Step 0, while the x-axis represents the current accuracy bin. **The scarcity of samples in the lower triangle demonstrates that performance degradation is rare.**

samples in each training batch and their respective proportions during the training process, as illustrated in Figure 12. It can be observed that approximately 40-60% of the actual training samples for BAPO come from online samples $\mathcal{X}_1$, while the remaining samples are derived from $\mathcal{X}_2$ or $\mathcal{X}_3$.

**Stability of Historical High-Quality Samples.** A potential concern regarding the reuse of historical high-quality samples ($\mathcal{X}_3$ in Eq. 5) is the assumption of policy consistency—specifically, whether samples that were high-quality under a past policy remain valid for the current policy. To address this, we visualize the evolution of sample difficulty in Figure 15 by tracking the accuracy migration of a training subset.

The heatmaps in Figure 15 reveal a distinct pattern: the mass is concentrated along the diagonal (performance maintenance) and the upper triangle (performance improvement). Crucially, the proportion of samples exhibiting significant performance degradation (migrating to the lower triangle) is negligible. For example, samples that initially achieved $8/8$ accuracy predominantly remain in the high-accuracy bins throughout the training process, with minimal regression to lower bins. This empirical evidence demonstrates that high-quality reasoning paths learned by RL are robust and resistant to forgetting. Consequently, historical high-quality samples stored in the buffer likely remain high-quality under the current policy, validating the consistency of the $\mathcal{X}_3$ data source.

A.7 HYPERPARAMETER SETTING

**Hyperparmeters** The major hyperparameter choices are shown in Table 3.

Table 3: **Hyperparameter Configuration** for BAPO Framework on Mathematics Task. For planning and visual geometry tasks, some parameters differ slightly; specific configuration scripts are provided in our code repository.

| Parameter | Value | Parameter | Value | Parameter | Value |
|---|---|---|---|---|---|
| **Rollout Configuration** | | | | | |
| Top-p | 1 | Top-k | -1 | Temperature | 1 |
| Group size ($G$) | 8 | Max prompt length | 2048 | Max response length | 8192 |
| Rollout workers | 8 | Sample batch size | 256 | Seed | 42 |
| **Training Configuration** | | | | | |
| Learning rate | 1e-6 | Train mini batch size | 64 | GAE lambda | 1.0 |
| Training epochs | 3 | KL coefficient ($\beta$) | 0.001 | Entropy coefficient | 0.001 |
| **Off-policy Configuration** | | | | | |
| $c_1$ threshold | 1/8 | $c_2$ range | [1/8, 4/8] | $c_3$ range | [2/8, 5/8] |
| Buffer size ($|B|$) | 256 | Rollout delay ($v$) | 5 | Re-evaluation freq ($m$) | 5 |
| Gaussian std ($\sigma$) | 0.2 | Gaussian mean ($\mu$) | 0.5 | Max re-evaluate prompts | 128 |

**Reward Function**  To evaluate the impact of our method, we adopt a simple reward function as below. All training experiments employ the same reward function.

$$r(x, y) = \begin{cases} 1, & \text{if } y \text{ is correct} \\ 0, & \text{otherwise} \end{cases}$$

**Datasets and Benchmarks**  To evaluate the models above, we use three training datasets and eight benchmarks categorized into mathematical, planning and vision geometry reasoning benchmarks as described in Table 4.

Table 4: **Datasets and Benchmarks** used in this study.

| Dataset | #Train | #Test | Task Type | Domain | License | Source |
|---|---|---|---|---|---|---|
| **Training Datasets** | | | | | | |
| DEEPSCALER-1.5B-PREVIEW | 40,000 | – | Math reasoning | Mathematics | Apache 2.0 | Link |
| COUNTDOWN-TASKS-3TO4 | 49,000 | – | Logic reasoning | Planning | Apache 2.0 | Link |
| GEOMETRY3K | 2,100 | – | Visual reasoning | Visual Geometry | Apache 2.0 | Link |
| **Test Benchmarks** | | | | | | |
| AIME24 | – | 30 | Math competition | Mathematics | MIT | Link |
| AMC | – | 83 | Math competition | Mathematics | Apache 2.0 | Link |
| MATH500 | – | 500 | Math reasoning | Mathematics | - | Link |
| MINERVA | – | 272 | Math reasoning | Mathematics | Apache 2.0 | Link |
| OLYMPIAD | – | 674 | Math competition | Mathematics | Apache 2.0 | Link |
| COUNTDOWN-TASKS-3TO4 | – | 200* | Logic reasoning | Planning | Apache 2.0 | Link |
| COUNTDOWN-TASKS-4 | – | 200* | Logic reasoning | Planning | Apache 2.0 | Link |
| GEOMETRY3K | – | 901 | Visual reasoning | Visual Geometry | Apache 2.0 | Link |

*We only use a random subset of this benchmark for faster ablation studies.

## A.8  ALGORITHM

Algorithm 1 presents the proposed BAPO, which can be seamlessly integrated with any GRPO-like RLVR algorithm.

---

**Algorithm 1** Batch Adaptation Policy Optimization (BAPO)

---

**Require:**  Policy $\pi_{\theta_0}$, buffer $\mathcal{B} = \emptyset$, thresholds $c_1, c_2, c_3$, delay steps $v$, re-evaluate frequency $m$
1: **for** $t = 1$ to $T$ **do**
2:     // Off-policy Rollout Phase
3:     **if** $t \bmod v = 0$ **then**
4:         Synchronize rollout policy's parameter with trainer: $\alpha = \pi_{\theta_t}$
5:     **end if**
6:     Using rollout policy $\alpha$ to generate $G$ responses $\{y_j\}_{j=1}^{G}$ for each question $x$
7:     Compute log probabilities $\alpha(y|x)$ and rewards $r$ for constructing the online batch $\mathcal{X}_{\text{on}}$
8:     Store samples into buffer $\mathcal{B}_{\text{bad}} \leftarrow \{(x, y, \alpha(y|x), r) \in \mathcal{X}_{\text{on}} : \mu_{\alpha,r}(x) \leq c_1\}$
9:     Store samples into buffer $\mathcal{B}_{\text{high}} \leftarrow \{(x, y, \alpha(y|x), r) \in \mathcal{X}_{\text{on}} : c_2 \leq \mu_{\alpha,r}(x) \leq c_3\}$
10:    // Off-policy Training Phase
11:    $\mathcal{X}_1 \leftarrow$ online filter on $\mathcal{X}_{\text{on}}$ with $\mu_{\alpha,r}(x) \in \{\frac{1}{G}, \dots, \frac{G-1}{G}\}$ (Filtered Fresh Samples)
12:    $\mathcal{X}_2 \leftarrow \emptyset$
13:    **if** $t \bmod m = 0$ **then**
14:       Re-evaluate $\mathcal{B}_{\text{bad}}$ with $\pi_{\theta_t}$ to get $\mathcal{X}_2$ using Equation 6 (Re-evaluated Difficult Samples)
15:    **end if**
16:    $\mathcal{X}_3 \leftarrow$ Sample from $\{(x, y) \in \mathcal{B}_{\text{high}} : \mu_{\alpha_{\mathcal{B}},r}(x) \in [c_2, c_3]\}$ (Historical High-quality Samples)

17:    Final batch $\leftarrow \mathcal{X}_1 \cup \mathcal{X}_2 \cup \mathcal{X}_3$
18:    Compute advantages and update critic/actor with final_batch
19:    Add $\mathcal{D}_t$ to buffer $\mathcal{B}$
20: **end for**

---

## A.9 GENERALIZATION ANALYSIS

To demonstrate the algorithmic generalizability of our framework, we extended the Batch Adaptation paradigm to Proximal Policy Optimization (PPO), denoted as **BA-PPO**. In this experiment, both the Actor and Critic networks were initialized with the **Qwen3-4B** backbone and trained on the **DeepScaleR** dataset with a maximum response length of 4K tokens. We maintained consistency with the foundational BAPO configuration by applying standard zero-advantage filtering for $\mathcal{X}_1$ (removing only all-correct and all-wrong groups), utilizing the initial BAPO values for thresholds $c_1, c_2, c_3$, and setting the buffer size to 64.

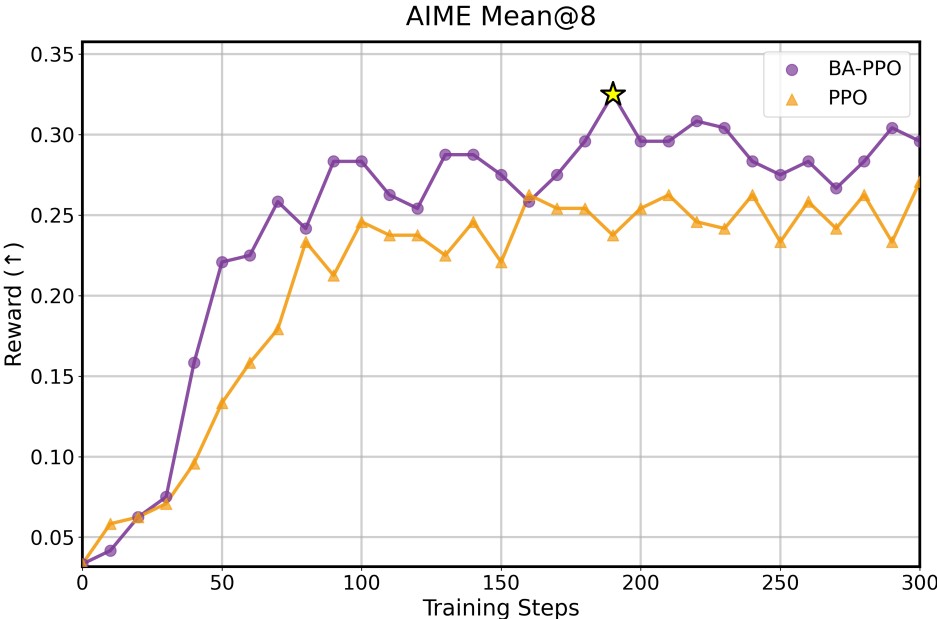

Figure 16: **Generalization to Actor-Critic Algorithms (BA-PPO).** Performance comparison between standard PPO (orange triangles) and BA-PPO (purple circles) on the AIME 2024 benchmark using Qwen3-4B. The star ($\star$) marks the peak performance of BA-PPO (0.325).

As illustrated in Figure 16, BA-PPO achieved a remarkable performance gain of **+5.5** on the **AIME 2024 benchmark** compared to the standard PPO baseline. This result further confirms that the core principle of dynamic batch construction is effective not only for GRPO but also functions as a robust, algorithm-agnostic enhancement for actor-critic methods.

