# OpenReview forum: "Buffer Matters: Unleashing the Power of Off-Policy Reinforcement Learning in Large Language Model Reasoning"
_ICLR.cc/2026/Conference — ICLR 2026 Poster_

### Official Review · Reviewer_x1w9 · 2025-10-28

**Soundness:** 2
**Presentation:** 2
**Contribution:** 2
**Rating:** 4
**Confidence:** 4

**Summary:**

This paper introduces Batch Adaptation Policy Optimization (BAPO), an off-policy reinforcement learning framework to improve the data efficiency of fine-tuning large language models on reasoning tasks. The core problem BAPO addresses is the failure of on-policy methods to learn effectively from difficult samples. BAPO constructs training batches by combining three data sources. It filters fresh rollouts to focus on moderately difficult samples. It also periodically re-evaluates historically difficult samples with the current policy to find newly solved ones. Finally, it reuses high-quality historical samples to ensure batch stability. Experiments show that BAPO improves performance over existing baselines.

**Strengths:**

1. The proposed batch construction method is well-motivated. Instead of simply mixing old and new data, it treats historical samples differently based on their difficulty. Re-evaluating hard examples and reusing successful ones is a more nuanced approach than standard experience replay and directly targets the stated problem of learning from difficult instances.

2. The evaluation in the paper covers three distinct reasoning domains (mathematics, planning, and visual geometry) using several modern LLM backbones. The comparison against a range of both on-policy and off-policy baselines is thorough, and the inclusion of detailed ablation studies strengthens the authors' claims about their design choices.

3. The work provides a solid theoretical foundation for the proposed method. The inclusion of a policy improvement lower bound in Theorem 3.2 offers a formal guarantee of training stability under certain conditions. This theoretical analysis elevates the work beyond a purely empirical contribution and builds confidence in the method's robustness.

**Weaknesses:**

1. The novelty of the core components could be overstated. The individual ideas of using a replay buffer, re-evaluating samples, and filtering data based on difficulty have precedents in the broader reinforcement learning literature, such as Prioritized Experience Replay. The main contribution lies in the specific combination of these techniques for LLM reasoning with verifiable rewards. The authors should further explain the novelty of this paper.

2. The mechanism for adapting the high-quality sample thresholds c2 and c3 is not clearly defined. The paper states that a linear mapping function is used based on the buffer's average performance, but the details of this function are omitted. This is a critical hyperparameter setting that could significantly impact performance, and its heuristic nature is not fully explored or justified with an ablation study.

**Questions:**

1. Regarding the adaptation of thresholds c2 and c3 mentioned in lines 248 to 251, could the authors provide the exact form of the linear mapping function used? How sensitive is the model's final performance to the specific design of this function?

2. Could the authors elaborate more on the novelty of BAPO in comparison to established methods? Specifically, what are the key distinctions in how BAPO re-evaluates and samples difficult historical data that make it particularly suitable for LLM reasoning tasks?

---

> ### Author Response · Authors · 2025-11-24
> **Response to Reviewer x1w9 [1/2]**
>
> We appreciate Reviewer x1w9's positive assessment of our motivation, the thorough evaluation across three distinct reasoning domains 1, and the solid theoretical foundation provided by Theorem 3.22. We now address the concerns regarding novelty and reproducibility:
>
> > **Q1: The novelty could be overstated; ideas like replay buffer and re-evaluation have precedents (e.g., PER). What makes BAPO distinct for LLMs?**
>
> **A1:** We acknowledge that Prioritized Experience Replay (PER) is a foundational concept. However, BAPO is not a direct application of PER; it is a specialized framework designed to address the unique challenges of **LLM RLVR**, specifically **reward sparsity** and **training stability**. The distinctions are fundamental:
>
> 1. **BAPO replaces the noisy TD-error metric with Verifiable Difficulty (Pass Rate) to effectively harness sparse rewards.** Unlike PER’s reliance on unstable TD errors in binary-reward settings, BAPO deterministically identifies "waste" data (historical failures) via the $\mathcal{X}_2$ mechanism. This allows us to convert zero-reward signal into valid learning signals through targeted re-evaluation.
>
> 2. **BAPO explicitly manages policy divergence through a Delayed Rollout mechanism to ensure training stability.** Unlike standard off-policy methods that simply sample batches from large volumes of historical data generated by disparate policies, BAPO enforces a constraint on the Total Variation (TV) distance between the buffer and the current policy. This design specifically prevents the "training collapse" often observed when reusing stale historical data for LLMs, ensuring that off-policy updates remain within a safe trust region.
>
> 3. **The "Mini-test" confirms that performance gains derive from the novel Hybrid Batch Architecture ($\mathcal{X}_1 + \mathcal{X}_2 + \mathcal{X}_3$), not heuristic tuning.** To prove that the novelty lies in this specific system design 66, we conducted a parameter-free "Mini-test" (Pages 8-9) where we stripped BAPO of all heuristic thresholds ($c_1, c_2, c_3$). We reduced the method to its theoretical core:
> 	- **$\mathcal{X}_1$ (Online):** Simple zero-advantage filtering (removing samples that are entirely correct or entirely wrong).
>
> 	- **$\mathcal{X}_2$ (Hard Replay):** Replay _only_ historically "all-wrong" samples (the exact samples filtered out by $\mathcal{X}_1$).
>
> 	- **$\mathcal{X}_3$ (High-Quality Reuse):** Reuse _only_ samples with exactly 50% accuracy ($p=0.5$). As proven in **Proposition A.3**, this probability theoretically maximizes the bound for single-step policy improvement.
>
>     **Conclusion:** Even when reduced to this minimal form, BAPO significantly outperforms the GRPO baseline. This confirms that the performance gains stem from the **structural paradigm** of re-evaluating hard samples and reusing high-variance historical samples to solve the exploration-stability trade-off, rather than hyperparameter engineering.

---

> ### Author Response · Authors · 2025-11-24
> **Response to Reviewer x1w9 [2/2]**
>
> > **Q2: The mechanism for adapting thresholds $c_2$ and $c_3$ is not clearly defined. How sensitive is the performance to this function?**
>
> **A2:** We apologize for the omission and have explicitly included the formulation in the revised **Section 3.2** to ensure reproducibility.
>
> 1. **Adaptive Linear Mapping Formula**
> 	We use a linear mapping to create a **"curriculum" effect**. Let $r_{tot}$ be the buffer's average reward (indicating model capability). As the model becomes stronger ($r_{tot}$ increases), we shift the focus toward harder samples (targeting a final range of $[1/8, 2/8]$) using the following formulas:
>
>     $$c_2 = r_{tot} \cdot (c_2^{high} - c_2^{low}) + c_2^{low}$$
>
>     $$c_3 = r_{tot} \cdot (c_3^{high} - c_3^{low}) + c_3^{low}$$
>
>     where $c^{high}$ and $c^{low}$ are the boundary hyperparameters detailed in the Appendix.
>
> 2. **Sensitivity Analysis**
> 	Regarding sensitivity, our ablation study and the **"Mini-test"** confirm that BAPO is **not brittle** with respect to threshold setting:
>
> 	- **Ablation Study (Figure 6, Column 3):** We conducted a detailed comparison that fixed the $\mathcal{X}_3$ reuse range to both **small values** (e.g., $[1/8, 2/8]$) and **large values** (e.g., $[4/8, 5/8]$), alongside the dynamic Linear Mapping. The experiments showed that even with fixed thresholds, BAPO consistently outperforms the GRPO baseline.
>
> 	- **"Mini-test" Validation:** The parameter-free "Mini-test" further validated this robustness by strictly fixing the reuse range to a single point ($p=0.5$).
>
> 	- **Conclusion:** The experimental results demonstrate that the adaptive function serves as an **optimization for efficiency and smoother convergence**, but the core performance gain is driven by the **architectural introduction** of the off-policy components $\mathcal{X}_2$ and $\mathcal{X}_3$, rather than the precise heuristic fine-tuning of this mapping function.
> ---
> Thanks for the attentive reading of the manuscript and constructive feedback. We have incorporated these changes into our revised version. We hope our response addresses all the concerns and that the reviewer will consider raising the rating accordingly. We are more than glad to answer any further questions.

---

> > ### Comment · Reviewer_x1w9 · 2025-11-27
> >
> > Thanks for your response. My concerns have been mostly resolved.
> >
> > While I still have minor questions about the computational trade-off of the re-evaluation step, I have increased my score. Maybe the authors can further explore it.

---

> > > ### Author Response · Authors · 2025-12-03
> > >
> > > We sincerely appreciate Reviewer x1w9's response and are glad to hear that your major concerns have been resolved. We are also grateful for your decision to **raise the score**.
> > >
> > > Regarding your remaining inquiry about the **computational trade-off of the re-evaluation step**, we conducted an additional ablation study to provide a more concrete analysis. To ensure a fair comparison, we utilized the exact same configuration as the **"Mini-test"** (Qwen3 8B, 4K length) described in our paper.
> > >
> > > We specifically ablated the **re-evaluation frequency ($m$)** and the **size of the bad case buffer ($\|B\|_{\text{bad}}$)**, as these two parameters directly determine the computational overhead of the re-evaluation phase. The results are summarized below:
> > >
> > > | Re-evaluate Ablations | Configuration | Accuracy(Mean@8) | Accuracy(Best@8) | Re-evaluate Total | X2 total |
> > > |----------------------|---------------|------------------|------------------|-------------------|----------|
> > > | **Frequency** | $m=2, \|B\|_{\text{bad}}=128$ | **0.354** | **0.528** | 200,480 | 65,824 |
> > > |  | $m=5, \|B\|_{\text{bad}}=128$ | 0.325 | 0.510 | 80,624 | 27,072 |
> > > |  | $m=10, \|B\|_{\text{bad}}=128$ | 0.312 | 0.470 | 40,304 | 14,176 |
> > > | **Buffer size** | $m=2, \|B\|_{\text{bad}}=128$ | 0.354 | 0.528 | 200,480 | 65,824 |
> > > |  | $m=2, \|B\|_{\text{bad}}=256$ | 0.333 | 0.500 | **392,064** | **81,984** |
> > >
> > > From these experiments, we derive two key insights regarding the trade-off:
> > >
> > > * **Insight 1: High-frequency re-evaluation yields better performance but with diminishing returns on efficiency.**
> > >     As shown in the table, increasing the re-evaluation frequency (lowering $m$ from 10 to 2) steadily improves the Mean@8 accuracy from 0.312 to 0.354. Meanwhile, the re-evaluate Total (number of inference calls) increases by roughly 5$\times$
> > >
> > > * **Insight 2: Larger bad-case buffers increase overhead without performance gains.**
> > >     Comparing the buffer sizes, expanding buffer size from 128 to 256 nearly doubles the re-evaluation cost and increases the total number of discovered valid samples ($\mathcal{X}\_2$ Total). However, interestingly, the downstream performance actually drops slightly from 0.354 to 0.333. This suggests that a smaller, fresher buffer ( $\|B\|_{\text{bad}}=128$ ) is sufficient to capture the most relevant "borderline" difficult samples. Excessive re-evaluation wastes sample utilization due to the limited training batch size.
> > >
> > > We hope this additional data clarifies the computational trade-offs involved in BAPO's design. We have optimized these parameters to ensure the method remains efficient while effectively recovering difficult samples.

---

### Official Review · Reviewer_Upr9 · 2025-10-31

**Soundness:** 2
**Presentation:** 2
**Contribution:** 3
**Rating:** 4
**Confidence:** 4

**Summary:**

This paper proposes a novel off-policy reinforcement learning framework called BAPO, which effectively addresses the issues of experience waste and reward homogeneity in traditional on-policy methods through an adaptive batch construction strategy that dynamically re-evaluates historically difficult samples and reuses high-quality ones. The paper provides a theoretical lower-bound guarantee and demonstrates outstanding performance and data efficiency across a series of mathematical, planning, and visual geometry reasoning tasks. Overall, the paper features a well-structured organization, well-justified motivations, and comprehensive experiments. However, several key design aspects in the methodology section are insufficiently explained, which affects the reproducibility and persuasiveness of the work.

**Strengths:**

1.	This paper proposes a method that systematically integrates the concept of off-policy reinforcement learning into the reinforcement learning with verifiable rewards for large language models. The method establishes a learning framework different from the traditional on-policy training paradigm by introducing experience replay and importance sampling mechanisms.

2.	The core innovation of the method lies in the design of a dynamic re-evaluation mechanism, which can reassess historical data according to the model's current capability level. This mechanism transforms previously underutilized training samples into valuable training data, thereby improving data efficiency.

3.	The adaptive batch construction strategy proposed in this paper manages the difficulty distribution of training data in a principled manner. By dynamically adjusting the proportion of samples of different difficulties, the method effectively avoids the problem of reward homogeneity that may occur during training.

4. Experiments conducted in the three major reasoning domains of mathematics, planning, and visual geometry, based on models of different scales, show that the method consistently outperforms a series of representative on-policy and off-policy baselines in comprehensive benchmark tests, empirically demonstrating its effectiveness.

**Weaknesses:**

1. **On the necessity of the Gaussian sampling in the $\mathcal{X}_1$ module.** The paper does not validate the performance impact of removing Gaussian sampling through ablation experiments, nor does it compare other sampling strategies (such as uniform sampling or difficulty-based sampling). Readers cannot determine whether this module is indispensable or merely a redundant design that increases methodological complexity. Therefore, it is recommended to supplement ablation experiments on Gaussian sampling and clarify the advantages of the Gaussian sampling strategy.

2. **Insufficient explanation of the $c_1/c_2/c_3$ threshold settings.** The values of $c_1/c_2/c_3$ in the paper lack adequate theoretical or experimental justification. Thresholds such as $c_1=0.125$, $c_2 \in [0,0.375]$, and $c_3 \in [0.25,0.5]$ appear arbitrary, with no explanation of their design inspiration or statistical basis. Moreover, although it is mentioned that $c_2$ and $c_3$ are based on "linear mapping," the specific mapping formula is not provided, making it impossible for readers to reproduce the adaptive process.

3. **Insufficient summarization of the principles behind the $c_1/c_2/c_3$ threshold values.** While the existing ablation experiments effectively demonstrate that "dynamic adjustment is useful," which is highly important, readers cannot derive a clear and transferable parameter design paradigm from them. The current ablation experiments resemble "effect validation" rather than "exploration of principles."

4. **Suggestion on terminology standardization and conceptual clarity.** The term "RFT" in the abstract is inconsistent with "RLVR" in the main text. Given that the method heavily relies on verifiable rewards, unifying the terminology to "RLVR" would enhance conceptual clarity and professionalism.

**Questions:**

Please refer to the “Weakness” section for related questions.

---

> ### Author Response · Authors · 2025-11-24
> **Response to Reviewer Upr9 [1/2]**
>
> We thank Reviewer Upr9 for recognizing the novelty of our dynamic re-evaluation mechanism and the principled nature of our adaptive batch construction. We now address the specific weaknesses raised:
>
> > **Q1: On the necessity of the Gaussian sampling in the $\mathcal{X}_1$ module. The paper does not validate its impact or compare with other strategies.**
>
> **A1:** We thank the reviewer for this constructive suggestion. To rigorously verify the necessity and robustness of our filtering logic, we implemented and compared three distinct online filtering modes in our ablation studies (detailed in **Appendix A.3**):
>
> - **Mode 1 (Range Filter):** Discards only zero-advantage samples (i.e., removing groups that are entirely correct or entirely wrong).
>
> - **Mode 2 (Gaussian Filter):** Assigns higher weight to samples with medium difficulty, reducing the proportion of extreme cases.
>
> - **Mode 3 (Uniform Filter):** Randomly selects 60% of fresh samples (matching the approximate retention rate of Modes 1 and 2) without considering difficulty.
>
>
> **Results:** Our experiments (**Figure 9**) indicate that **Mode 3 (Uniform)** leads to training instability, confirming that a value-based filtering logic is necessary to filter out noise. However, the performance difference between **Mode 1 (Range)** and **Mode 2 (Gaussian)** is task-dependent. While Gaussian filtering aids tasks with buffer initial performance, Range filtering is generally sufficient.
>
> **Revision Action:** We have moved the detailed analysis of $\mathcal{X}_1$ to **Appendix A.3** to focus the main text on our core, novel off-policy contributions: **Historical Difficult Sample Replay ($\mathcal{X}_2$)** and **Historical High-Quality Sample Reuse ($\mathcal{X}_3$)**.
>
> > **Q2: Insufficient explanation of the $c_1/c_2/c_3$ threshold settings and the missing mapping formula.**
>
> **A2:** We apologize for the omission and have explicitly defined these in the revised **Section 3.2**.
>
> First, we clarify that the group size is set to $G=8$. Since individual rewards are binary $r \in \{0, 1\}$, the group mean reward (Pass Rate) can only take discrete values from the set $\{0, 1/8, 2/8, \dots, 7/8, 1\}$. We have updated the manuscript to clearly articulate these definitions. Based on this, our thresholds are selected as follows:
>
> 1. Theoretical Basis:
>
>     - **$c_1$ (Difficulty Threshold):** As illustrated in **Figure 15** (Transition Matrix of Sample Difficulty), statistical evidence from training dynamics shows that samples initially labeled "All Wrong" ($0/8$) or "Correct Once" ($1/8$) exhibit a much higher probability of stagnation or degradation compared to easier samples. This identifies them as "hard samples" that possess high training value if solved.
>
>     - **$c_2, c_3$ (High-Quality Thresholds):** These are initialized around the theoretical optimum of **0.5** (e.g., $[4/8, 5/8]$). As proven in **Proposition A.3**, samples with a pass rate of $p=0.5$ provide the **theoretical maximum gradient variance** for single-step policy improvement ($J(\pi_\theta) - J(\pi_{\theta_t})$).
>
> 2. Adaptive Linear Mapping Formula:
>
>     To address reproducibility, we provide the exact linear mapping formulas used to shift the difficulty window as the model improves. Let $r_{tot}$ be the buffer's average reward. As the model becomes stronger ($r_{tot}$ increases), we shift the focus toward harder samples (targeting a final range of $[1/8, 2/8]$) to maintain a "curriculum" effect:
>
>     $$c_2 = r_{tot} \cdot (c_2^{high} - c_2^{low}) + c_2^{low}$$
>
>     $$c_3 = r_{tot} \cdot (c_3^{high} - c_3^{low}) + c_3^{low}$$
>
>     Where $c^{high}$ and $c^{low}$ are the boundary hyperparameters detailed in the Appendix.
>
>
> **Crucially**, as demonstrated by the ablation studies on thresholds $c_1-c_3$ (**Figure 6**) and the **"Mini-test"** (detailed below), BAPO's performance gains derive primarily from the **architectural introduction of the off-policy components $\mathcal{X}_2$ and $\mathcal{X}_3$**, rather than the heuristic fine-tuning of these threshold parameters.

---

> ### Author Response · Authors · 2025-11-24
> **Response to Reviewer Upr9 [2/2]**
>
> > **Q3:  Insufficient summarization of the principles behind the $c_1/c_2/c_3$ threshold values... The current ablation experiments resemble "effect validation" rather than "exploration of principles."**
>
> **A3:** We agree that principles are more important than parameters. To address this, we introduced a **"Mini-test" (Parameter-Free Validation)** in the revised manuscript (Pages 8-9) to validate the transferable design paradigm.
>
> In this test, we reduced BAPO to its theoretical core without any heuristic parameter tuning:
>
> - **$\mathcal{X}_1$ (Online):** Apply standard zero-advantage filtering only (removing all-correct/all-wrong samples). This is a universal principle requiring no threshold tuning.
>
> - **$\mathcal{X}_2$ (Hard Replay):** Replay _only_ historical all-wrong samples (corresponding to $c_1=0$). This targets the "waste" data identified by $\mathcal{X}_1$ without arbitrary thresholds.
>
> - **$\mathcal{X}_3$ (High-Quality Reuse):** Reuse _only_ samples with exactly **50% accuracy**. This is based strictly on the theoretical proof that $p=0.5$ yields maximum gradient variance.
>
> **Conclusion:** Even in this restricted "Mini-test," BAPO significantly outperforms the GRPO baseline. This confirms that the performance gains stem from the **structural paradigm** of re-evaluating hard samples and reusing high-variance historical samples, rather than hyperparameter engineering. This conclusion is further supported by our general ablation study (**Figure 6**), which demonstrates that while BAPO is robust to parameter variations, removing the structural components ($\mathcal{X}_2$ or $\mathcal{X}_3$) leads to immediate performance degradation. We have included these results to provide a clear, transferable design paradigm for future research.
>
> > **Q4: Suggestion on terminology standardization and conceptual clarity. The term "RFT" in the abstract is inconsistent with "RLVR" in the main text. Given that the method heavily relies on verifiable rewards, unifying the terminology to "RLVR" would enhance conceptual clarity and professionalism.**
>
> **A4:** We value the suggestion on terminology and have unified it to "**RLVR**" (Reinforcement Learning with Verifiable Rewards) throughout the revision to enhance conceptual clarity.
>
> ---
> Thanks for the attentive reading of the manuscript and constructive feedback. We have incorporated these changes into our revised version. We hope our response addresses all the concerns and that the reviewer will consider raising the rating accordingly. We are more than glad to answer any further questions.

---

### Official Review · Reviewer_SdcC · 2025-10-31

**Soundness:** 3
**Presentation:** 3
**Contribution:** 2
**Rating:** 6
**Confidence:** 4

**Summary:**

The paper proposes a general sample optimization algorithm that ingeniously integrates both on-policy and off-policy data to train the policy model in a targeted manner, adaptively selecting samples based on their learning difficulty assessed from rollouts.

**Strengths:**

- The paper clearly identifies the key limitations of existing on-policy RLVR methods (experience waste and reward homogeneity) and proposes a well-motivated off-policy framework with intuitive design principles that align with RL fundamentals.
- The paper provides rigorous theoretical analysis (Theorem 3.2) proving that the adaptive batch construction mechanism maintains a lower bound guarantee for policy improvement, ensuring training stability while leveraging off-policy data.
- The evaluation across diverse reasoning tasks and model backbones demonstrates consistent improvements over baselines, with thorough ablation studies confirming the effectiveness of the proposed components.

**Weaknesses:**

- While BAPO effectively improves learning efficiency through adaptive sample selection, it primarily reorganizes existing experiences rather than fundamentally expanding the model's exploration space. This may limit its ability to solve problems that consistently fail under the current policy distribution.
- The paper primarily compares rollout counts, but lacks detailed analysis of the actual computational overhead, particularly the costs of forward passes (e.g., computing log probabilities for importance sampling ratios) and backward passes during training. Since BAPO reuses historical samples that do not require new rollouts but still incur training costs, a comprehensive breakdown of these components would better clarify the true efficiency gains.

**Questions:**

- BAPO is specifically designed for GRPO-like group-based advantage estimation, where homogeneous rewards within a group can lead to zero gradients. It remains unclear whether this framework generalizes to other on-policy RL algorithms such as PPO or Reinforce, which does not suffer from the same gradient vanishing issues.
- Does BAPO incorporate any mechanisms to encourage exploration beyond the existing experience pool, or is performance ultimately bounded by the policy's rollout coverage?
- BAPO w/o X3 shows a substantial performance gap compared to full BAPO. Since X3 only provides historical high-quality samples that have already been used, could you explain the source of this large gap? Is this primarily due to the mismatch in effective batch sizes (as you configured BAPO w/o X3 to retain all fresh samples X1), or does the historical data contribute beyond simply maintaining batch size consistency?

---

> ### Author Response · Authors · 2025-11-24
> **Response to Reviewer SdcC [1/3]**
>
> We are pleased that Reviewer SdcC finds our motivation well-grounded and our method reasonable, and appreciates the rigor of our theoretical analysis and the thoroughness of our ablation experiments. Below, we address the reviewer's specific questions:
>
> > **Q1: While BAPO effectively improves learning efficiency through adaptive sample selection, it primarily reorganizes existing experiences rather than fundamentally expanding the model's exploration space...Does BAPO incorporate any mechanisms to encourage exploration beyond the existing experience pool?**
>
> **A1:** We appreciate this insightful observation. We clarify that BAPO is not designed as an active exploration algorithm (i.e., it does not force the model into entirely unseen semantic spaces via entropy regularization or tree search). However, BAPO significantly enhances the **exploitation of the exploration frontier**, effectively converting **"failed exploration"** into **"successful learning"** through two mechanisms:
>
> 1. **Re-evaluation as a "Multi-Chance Strategy":**
>
>     - Existing on-policy methods (e.g., DAPO [1]) typically discard failed rollouts immediately to maintain training batch quality, leading to a high rollout burden. Similarly, while approaches like Knapsack RL [2] increase rollouts for hard samples, they are still limited to using the current, single policy for difficult problems.
>
>     - BAPO stores these **"exploration failures"** and periodically re-evaluates them. This is equivalent to assessing a hard sample with **multiple future policies** across different training stages, significantly increasing the probability that the sample will be solved. As the policy improves, these previously failed explorations often turn into successful trajectories. By capturing this transition moment immediately via $\mathcal{X}_2$, we maximize the utility of the model's natural exploration.
>
> 2. **Unlocking "Borderline" Samples via Sustained Exposure:**
>     - BAPO continuously reuses **samples with moderate accuracy (FIFO to control freshness)** that match the model's current capability curve. By focusing updates on these high-variance "borderline" cases rather than fully mastered or completely impossible ones, BAPO forces the model to consolidate its knowledge at the exploration frontier.
>     - As shown in **Figure 7**, we tracking the accuracy change of each samples before and after RLVR. BAPO successfully unlocks **31%** of samples that were initially unsolvable ($0/G$ accuracy), compared to only 19% for GRPO. This indicates that BAPO allows the model to master problems on the "boundary" of its capability much faster than standard baselines, pushing the limits of the current exploration space.
> ---
> [1] Q Yu, et, al. "DAPO: An Open-Source LLM Reinforcement Learning System at Scale." NeurIPS2025
>
> [2] Z Li, et, al. "Knapsack RL: Unlocking Exploration of LLMs via Optimizing Budget Allocation" arxiv

---

> ### Author Response · Authors · 2025-11-24
> **Response to Reviewer SdcC [2/3]**
>
> > **Q2: The paper primarily compares rollout counts, but lacks detailed analysis of the actual computational overhead, particularly the costs of forward passes (e.g., computing log probabilities for importance sampling ratios) and backward passes during training. Since BAPO reuses historical samples that do not require new rollouts but still incur training costs, a comprehensive breakdown of these components would better clarify the true efficiency gains.**
>
> **A2:** We thank the reviewer for requesting this crucial transparency. The efficiency of BAPO relies on a calculated trade-off between **"extra forward passes"** (re-evaluation) and **"reduced backward passes"** (filtering):
>
> 1. **Forward Pass Overhead (Cost Increase):** BAPO indeed incurs additional forward pass costs for re-evaluating $\mathcal{X}_2$ samples. However, as demonstrated in **Figure 12 and Table 1**, BAPO substantially reduces the overall total rollout count compared to baseline methods like DAPO while achieving comparable or superior performance on test benchmarks. For example, on the mathematics task, BAPO consumes **733k rollouts**, representing only an **8% increase** compared to the baseline GRPO's **677k rollouts**, but a **62% reduction** compared to DAPO's **1,921k rollouts** (which uses dynamic sampling to maintain training batch valuable). The $\mathcal{X}_2$ re-evaluation cost is efficiently managed within the buffer's capacity limits.
>
> 2. **Backward Pass Savings (Cost Decrease):** Crucially, BAPO filters out low-information online samples ($\mathcal{X}_1$ filtering of zero-variance data). As illustrated in the new **Figure 8** (which shows the quantity and source of samples in the training batch) and **Figure 14** (Rollout count comparison), the filtering significantly affects the backward pass. The **effective batch size** for BAPO’s backward propagation is often smaller than the maximum capacity when training on the high-information $\mathcal{X}_2$ and $\mathcal{X}_3$ subsets.
>
> 3. **Net Efficiency:** Since backward propagation is computationally expensive, the savings from the smaller, high-quality effective batch size largely offset the overhead of re-evaluation. This explains why BAPO achieves training times comparable to GRPO (as shown in **Table 2**) while being substantially faster and more data-efficient than massive-rollout methods like DAPO.
>
> > **Q3: BAPO is specifically designed for GRPO-like group-based advantage estimation... It remains unclear whether this framework generalizes to other on-policy RL algorithms such as PPO or Reinforce.**
>
> **A3:** Thank you very much for your valuable suggestion. While our original implementation focused on GRPO, the core principles of BAPO are algorithm-agnostic and theoretically transferable to PPO or REINFORCE-style algorithms.
>
> - The fundamental novelty of BAPO lies in the **historical sample utilization ($\mathcal{X}_2$ and $\mathcal{X}_3$),** not the advantage estimation method. The criteria for sample selection (pass rates $c_1, c_2, c_3$) can be realized regardless of whether the advantage is estimated using a Critic (PPO) or Group Mean Reward (GRPO).
>
> - To validate this generalizability, we have **added a comparative experiment** using **PPO** and **BA-PPO** (Batch Adapted PPO). We used Qwen3 4B as the actor and critic, with all other batch adaptation parameters being consistent with the GRPO-style setup.
>
> The results, detailed in the **Appendix** A.9, show that **BA-PPO converges faster and achieves +5.5  performance gain than standard PPO** on the AIME 2024 benchmark. This confirms that BAPO's structural innovation is a generalizable paradigm for improving data efficiency in RLVR.

---

> ### Author Response · Authors · 2025-11-24
> **Response to Reviewer SdcC [3/3]**
>
> > **Q4: BAPO w/o $\mathcal{X}_3$ shows a substantial performance gap compared to full BAPO... Is this primarily due to the mismatch in effective batch sizes... or does the historical data contribute beyond simply maintaining batch size consistency?**
>
> **A4:** Thank you very much for your insightful question. The performance gap caused by removing $\mathcal{X}_3$ is primarily due to the **high informational quality** of these samples, which contributes to gradient stability far beyond simply maintaining the batch size.
>
> 1. **$\mathcal{X}_3$ provide samples that maximize the learning potential for the current model, specifically by targeting the theoretical optimum for policy improvement.**
>
>    In BAPO, $\mathcal{X}_3$ samples are specifically selected to have **high variance** or high learning potential. As formally proven in **Proposition A3.2**, samples around this $\mu=0.5$ threshold offer the **tightest lower bound for policy improvement**. These high-variance samples are highly valuable training targets because they represent the region where the model is most uncertain, making the resulting gradient signal most informative.
>
> 2. **$\mathcal{X}_3$ acts as a high-quality "anchor" that stabilizes the gradient direction and prevents oscillation, ensuring reliable maximization of the improvement lower bound.**
>
>    If we remove $\mathcal{X}_3$, the training batch relies solely on $\mathcal{X}_1$ (online samples) and $\mathcal{X}_2$ (hard solved samples). These remaining subsets can be noisy or unstable (e.g., all-correct or all-incorrect samples in $\mathcal{X}_1$; $\mathcal{X}_2$ samples are often "barely solved" and may yield volatile gradient estimates). By incorporating data known to yield high-leverage, stable gradients, $\mathcal{X}_3$ ensures that the overall policy update step reliably maximizes the lower bound on improvement, leading to smoother and higher convergence compared to the volatile trajectory of BAPO w/o $\mathcal{X}_3$.
> ---
> Thanks for the attentive reading of the manuscript and constructive feedback. We have incorporated these changes into our revised version. We hope our response addresses all the concerns and that the reviewer will consider raising the rating accordingly. We are more than glad to answer any further questions.

---

> > ### Comment · Reviewer_SdcC · 2025-11-25
> >
> > I thank the authors for their reponse. The additional clarifications have addressed most of my concerns. I will leave my score as it is, as my review was mostly asking for additional clarifications.

---

### Official Review · Reviewer_g1rK · 2025-11-01

**Soundness:** 2
**Presentation:** 2
**Contribution:** 2
**Rating:** 4
**Confidence:** 4

**Summary:**

The paper introduces Batch Adaptation Policy Optimization (BAPO), an off-policy reinforcement learning fine-tuning framework for LLM reasoning tasks. It incorporates two key off-policy designs into standard RL post-training framework: (1) during rollout process, it delays the update of the inference server to mitigate rollout policy instability that may arise from rapid distribution shifts (a technique from prior work), and (2) during training process, BAPO dynamicaly constructs training batches that combine online generated samples, re-evaluated difficult samples, and historical high-quality samples. Experiments on mathematical reasoning benchmarks demonstrate that BAPO achieves higher accuracy compared to baseline methods.

**Strengths:**

- The ablation study is comprehensive, including ablations on $\mathcal{X}_2, \mathcal{X}_3$, delay steps (v), re-rollout frequency (m), and difficulty thresholds $c_1,c_2,c_3$.
- Experiments span multiple reasoning domains, demonstrating versatility.
- The paper structure is clear.

**Weaknesses:**

- Overall, the method appears to be a collection of empirical tricks rather than a broadly generalizable or methodologically novel approach. Specifically, The method introduces a large number of hyperparameters and custom design choices—such as $c_1, c_2, c_3$,  online sample mean ($\mu$) and standard deviation ($\sigma$), re-rollout frequency (m), linear mapping function, rollout delay steps (v), and the proportion of different sample types, which greatly limit its plug-and-play usability.
- The overall method design is highly heuristic, with limited justification on its rational. For example, in Eq. (5), the use of “historical high-quality samples” assumes that samples that were once high-quality remain so under the current policy. Although the method designs a linear mapping function to determine $(c_2, c_3)$ based on the buffer’s average performance, this approach is coarse and lacks solid reasoning to ensure that the selected samples are indeed high-quality under the updated policy.
- In the $\mathcal{X}_2$ filtering phase, the method requires periodically re-generating responses for all historically difficult samples using the current policy, leading to substantial computational overhead—especially when the dataset contains numerous difficult samples (a common scenerio). Similarly, in the $\mathcal{X}_1$ filtering phase, not all online rollouts are utilized, resulting in additional inefficiency.
- The paper lacks details on the quantities of $|\mathcal{X}_1|, |\mathcal{X}_2|, |\mathcal{X}_3|$, which are crucial. Specifically, for $\mathcal{X}_1$, how many online rollouts are retained out of the total generated? For $\mathcal{X}_2$, since the number of such samples is varies considerably, how are the sample sizes distributed and balanced across the three sample types?
- The theoretical proofs rely on assumptions of small total variation distance, but these assumptions are not sufficiently validated. In particular, assuming $\mathrm{TV}(\pi_{\theta_t}(\cdot | x), \alpha_B(\cdot | x)) \le \delta_3$ with sufficiently small $\delta_3$ could severely limit the improvement of $\pi_{\theta_t}$.

**Questions:**

- In the computation analysis results (Line 457), it is counter-intuitive that BAPO is not slower than GRPO, which requires clarification. Given identical training batch sizes, the generation cost before online rollout filtering should be similar to that of GRPO (assuming the pre-filtering sample size is the same, which is not described in the paper). Moreover, constructing $|\mathcal{X}_2|$ requires periodic large-scale computation of the current policy probabilities over all historically difficult samples. And during training, off-policy samples also require additional probability computation under the current policy.

---

> ### Author Response · Authors · 2025-11-24
> **Response to Reviewer g1rK [1/2]**
>
> We are glad that Reviewer g1rK found our ablation studies comprehensive and appreciated the versatility of our experiments across multiple reasoning domains. We now address the reviewer's specific questions below:
>
> > **Q1: The method appears to be a collection of empirical tricks with many hyperparameters ($c_1, c_2, c_3$, etc.), limiting plug-and-play usability.**
>
> **A1:** We appreciate this feedback and the opportunity to verify the robustness of our design. We would like to clarify that our core contribution is the **systematic framework** for utilizing stale off-policy data, rather than the specific hyperparameters. To demonstrate this, we provide two lines of evidence:
>
> 1. Structural Validation via "Mini-test"
>
> To prove that performance stems from the framework design rather than parameter tuning, we conducted a "Mini-test" (Parameter-Free Validation) in the revised manuscript (Pages 8-9). In this experiment, we removed all heuristic thresholds ($c_1, c_2, c_3$) and used only raw theoretical principles:
>
> - **$\mathcal{X}_1$ (Online):** Simple zero-advantage filtering (removing samples that are entirely correct or entirely wrong).
>
> - **$\mathcal{X}_2$ (Hard Replay):** Replay _only_ historically "all-wrong" samples (exactly matching the difficult cases discarded by $\mathcal{X}_1$).
>
> - **$\mathcal{X}_3$ (High-Quality Reuse):** Reuse _only_ samples with exactly **50% accuracy** ($p=0.5$). As proven in **Proposition A.3**, this probability theoretically maximizes the lower bound for single-step policy improvement.
>
>
> **Experiment Setting & Results:** We aligned all settings with GRPO and compared against a controlled **DAPO** baseline that used the **exact same $\mathcal{X}_1$ filter**. As shown in **Figure 5**, even this hyperparameter-free BAPO significantly outperforms both GRPO and the controlled DAPO baseline. This confirms that the gains stem from the **structural innovation** (re-evaluating hard samples and reusing high-quality ones), not parameter engineering. Furthermore, our ablation study (**Figure 6, Columns 1 & 3**) demonstrates that removing these structural components leads to immediate performance degradation, while variations in parameters have minimal impact.
>
> 2. Plug-and-Play Implementation
>
> To ensure practical "plug-and-play" usability, we have fully decoupled all functional modules in our code implementation. Users can independently enable or disable specific components via simple configuration flags without complex setup. For example:
>
> - `enable_off_policy_rollout`: Enable the delayed rollout policy mechanism ($\pi_{t-v}$).
>
> - `enable_off_policy_reeval`: Controls the re-evaluation of historical hard samples ($\mathcal{X}_2$).
>
> - `enable_completion_batch`: Manages the reuse of historical high-quality samples ($\mathcal{X}_3$).
>
>
> This modular design ensures that BAPO can be easily integrated into existing pipelines or customized for specific resource constraints with minimal effort.
>
> > **Q2:  The overall method design is highly heuristic, with limited justification on its rational. For example, in Eq. (5), the use of “historical high-quality samples” assumes that samples that were once high-quality remain so under the current policy.**
>
> **A2:** We thank the reviewer for challenging the rationale behind reusing historical samples. We explicitly address this concern in the revised manuscript under the section **"Stability of Historical High-Quality Samples"** and provide empirical validation in **Figure 15** (Appendix A.6).
>
> - **Experimental Setup:** To verify this assumption, we tracked the evolution of sample difficulty throughout the training process. We categorized samples into difficulty bins based on their initial accuracy (e.g., $0, \frac{1}{G}, \dots, \frac{G-1}{G}, 1$) and recorded how these samples' accuracy shifted between policy checkpoints.
>
> - **Empirical Verification (Figure 15):** We visualized these dynamics using an **Accuracy Transition Matrix**, which maps the probability of a training sample migrating from one accuracy interval to another as the policy updates. The results reveal a distinct pattern: the probability mass is heavily concentrated along the **diagonal** (indicating performance maintenance) and the **upper triangle** (indicating performance improvement).
>
> - **Conclusion:** Crucially, we observe that as sample accuracy improves, the probability of regression to a "Difficult" state (e.g., zero or near-zero accuracy) diminishes significantly. This empirically validates the assumption in Eq. (5): historical high-quality samples retain their validity under the updated policy, allowing for efficient reuse without the computational cost of re-evaluation.

---

> ### Author Response · Authors · 2025-11-24
> **Response to Reviewer g1rK [2/2]**
>
> > **Q3: In the $\mathcal{X}_2$ filtering phase, the method requires periodically re-generating responses for all historically difficult samples using the current policy... The paper lacks details on the quantities of $\mathcal{X}_1,\mathcal{X}_2,\mathcal{X}_3$, which are crucial....In the computation analysis results (Line 457), it is counter-intuitive that BAPO is not slower than GRPO, which requires clarification.**
>
> **A3:** We thank the reviewer for this constructive suggestion. We have added a new subsection "**Sample Distribution & Efficiency**" to clarify the computational dynamics. BAPO achieves efficiency through the following mechanisms:
>
> 1. **Controlled Overhead (FIFO) for $\mathcal{B}\_{\text{bad}}$ and $\mathcal{B}\_{\text{high}}$:** As shown in **Table 3**, the re-evaluation volume for $\mathcal{X}_2$ is strictly limited by the `buffer_batch_size` (aligned with the training batch size) using a First-In-First-Out (FIFO) mechanism. We do _not_ re-generate **all historical samples**, keeping the inference cost controllable. We have also updated the **Method** section to clarify that $\mathcal{X}_2$ and $\mathcal{X}_3$ employ the FIFO strategy to manage buffer size.
>
> 2. **Gradient Cost-Effectiveness ($\mathcal{X}_1$):** Discarding zero-variance online samples is a deliberate choice. These samples provide near-zero gradient signals but consume backward-pass computation. To further justify our choice, we added a comparison of three filtering strategies (Uniform, Range, Gaussian) in **Appendix A.3**,  our experiments (**Figure 9**) indicate that **Uniform filter** leads to training instability, confirming that a value-based filtering logic is necessary to filter out noise samples. However, the performance difference between **Range filter** and **Gaussian filter** is task-dependent. While Gaussian filtering aids tasks with buffer initial performance, Range filtering is generally sufficient. **Since online filtering is not the primary focus of this paper, we have moved this analysis to Appendix A.3 and revised Section 3.2 accordingly.**
>
> 3. **Dynamic Speed Balance:** As illustrated in **Figure 8**, the total batch size $|\mathcal{X}_1| + |\mathcal{X}_2| + |\mathcal{X}_3|$ does not always fill the maximum `train_batch_size`, particularly in the early training stages (first ~50 steps) where solved hard / high-quality samples are scarce. Consequently, the time saved during the **backward pass** (due to smaller effective batch sizes) effectively offsets the additional **inference time** required for re-evaluation.
>
>     - _Note:_ While this balance typically maintains speed parity, there are exceptions. For the **Mathematics** task (**Table 2**), BAPO is slightly slower than GRPO due to the high volume of valid samples, but it remains significantly faster than DAPO while achieving superior test scores.
>
>
> > **Q4: The theoretical proofs rely on assumptions of small total variation (TV) distance, which may limit improvement.**
>
> **A4:** We thank the reviewer for this insightful question. We argue that the small TV distance is not merely a strict assumption, but rather a practical condition that can be controlled through our design. We address this from three perspectives:
>
> 1. **Trust Region ($\mathcal{X}_1$):** The inherent Trust Region nature of GRPO/PPO-style algorithms guarantees the stability of the short-term policy $\pi_{\theta_t}$, bounding the divergence for online samples.
>
> 2. **FIFO Buffer ($\mathcal{X}_3$):** The First-In-First-Out mechanism ensures the "freshness" of the buffer policy $\alpha_B$, bounding its divergence from the current policy.
>
> 3. **Scope of Constraint:** Crucially, the TV constraint mainly applies to $\mathcal{X}_3$, which constitutes only ~20-40% of the data. Therefore, the model retains the capacity for significant improvement via fresh policy's rollouts.
> ---
> Thanks for the attentive reading of the manuscript and constructive feedback. We have incorporated these changes into our revised version. We hope our response addresses all the concerns and that the reviewer will consider raising the rating accordingly. We are more than glad to answer any further questions.

---

### Author Response · Authors · 2025-11-24
**General response to all reviewers**

Dear PC, SAC, AC, and all reviewers,

We sincerely thank you for all the reviewers' thoughtful reviews and invaluable feedback. We truly appreciate the time and effort you devoted to carefully reading our paper. Your insights have been extremely helpful in strengthening the clarity, rigor, and breadth of our work.

We are grateful that the reviewers recognized several positive aspects of **BAPO**, including its "**well-motivated off-policy framework**" (Reviewer SdcC, x1w9), "**comprehensive experiments across multiple domains**" (Reviewer g1rK, SdcC, Upr9, x1w9), and "**clear structure and presentation**" (Reviewer g1rK, Upr9).

To address the comments and concerns raised, we have prepared individual, point-by-point responses for each reviewer. We have also extensively revised the manuscript, adding **5 new figures**, **1 new main text section**, and **4 new appendix sections**. A summary of the key modifications follows below:

**1. Validation of Structural Novelty (Addressing "Heuristics"):**

- **Added "Mini-test" (Parameter-Free Validation):** To address concerns about hyperparameter dependency (Reviewer g1rK, Upr9), we introduced a "Mini-test" (Pages 8-9, **Figure 5**). This experiment strips BAPO of all heuristic thresholds and uses only raw theoretical principles ($\mathcal{X}_1$: Zero-Advantage Filter; $\mathcal{X}_2$: All-Wrong Replay; $\mathcal{X}_3$: $\mu=0.5$ Reuse). The results confirm that BAPO's performance stems from its **structural design**, not parameter tuning.

- **Added Theoretical Basis for Thresholds:** We added **Proposition A.3** to prove that samples with a accuracy of $\mu=0.5$ maximize the gradient variance for single-step policy improvement, providing a theoretical foundation for our high-quality sample selection.


**2. Efficiency & Computational Cost Analysis:**

- **Dynamic Batch Analysis:** We added **Figure 8** and revised **Appendix A.5** to visualize the dynamic composition of the training batch. This demonstrates that BAPO often trains with a reduced effective batch size (filtering out zero-information online samples), which offsets the computational cost of re-evaluation.

- **Rollout Cost Comparison:** We added **Figure 12** to explicitly compare cumulative rollout batches, showing that BAPO requires significantly fewer rollouts than heavy-sampling methods like DAPO while achieving superior performance.


**3. Mechanism Verification & Generalizability:**

- **Stability of Historical Samples:** To address concerns about the validity of reusing old data (Reviewer g1rK), we revised **Appendix A.6** and added **Figure 15**. This empirically proves that "High-Quality" samples rarely degrade back to "Hard" states, validating the stability of the $\mathcal{X}_3$ buffer.

- **Generalization to Actor-Critic (BA-PPO):** We added **Appendix A.7** and **Figure 16**, extending the Batch Adaptation paradigm to **PPO**. The results show that BA-PPO significantly outperforms standard PPO, confirming that our framework is algorithm-agnostic.

- **Online Filter Ablation:** We added **Appendix A.3** and **Figure 9** to compare Range, Gaussian, and Uniform filtering, justifying the necessity of our value-based filtering strategy.


**4. Clarity & Reproducibility:**

- **Explicit Formulas:** We added the exact linear mapping formulas for adaptive thresholds $c_2$ and $c_3$ in **Section 3.2**.

- **Unified Terminology:** We standardized the terminology to "**RLVR**" (Reinforcement Learning with Verifiable Rewards) throughout the paper.


We hope that our detailed responses and the substantial revisions to the paper satisfactorily address your concerns. We look forward to any further discussion.

---

### Author Response · Authors · 2025-12-04
**Summary of Reviewer Feedback and Our Responses**

Dear Program Chairs, Senior Area Chairs, Area Chairs, and Reviewers,

We sincerely thank you for your time and efforts during the review process, and we extend our gratitude to all reviewers for their constructive feedback.

In response to the reviewers’ questions and suggestions, we have added **5 new figures**, **1 new main text section**, and **4 new appendix sections**, providing a more comprehensive analysis and evaluation of our method. All modifications in the revised paper are highlighted in blue.

We are pleased to note that **two reviewers (SdcC and x1w9) acknowledged that most of their concerns have been addressed**, with Reviewer x1w9 explicitly **raising their score (from 4 to 6)** in recognition of the improvements made.

Below, we summarize the key concerns raised during the review process and explain how we have addressed them. For completeness, detailed point-by-point responses to each reviewer are also provided in the individual replies.

---

## 1. **Methodological novelty and hyperparameter sensitivity**
**Raised by:** Reviewers g1rK and Upr9 expressed questioned its sensitivity to hyperparameters.

**Addressed by:** We introduced a **"Mini-test" (Parameter-Free Validation)** (Pages 8–9, Figure 5) that strips BAPO of all heuristic thresholds and uses only core theoretical principles:
- $\mathcal{X}\_1$: Zero-advantage filtering
- $\mathcal{X}\_2$: All-wrong replay
- $\mathcal{X}\_3$: Reuse of samples with exactly 50% accuracy

The results confirm BAPO's performance stems from its structural design, not parameter tuning. We also added **Proposition A.3** to provide theoretical grounding for the $\mu = 0.5$ threshold, which maximizes gradient variance for single-step policy improvement.

---

## 2. **Computational overhead and efficiency analysis**
**Raised by:** Reviewers g1rK and SdcC requested a clearer analysis of computational costs and efficiency trade-offs.

**Addressed by:** We added a detailed computational cost analysis in **Figure 12** and **Appendix A.5**, comparing rollout consumption across baselines. The results show BAPO significantly reduces rollout requirements while maintaining superior performance. We also visualized the dynamic batch composition in **Figure 8**, demonstrating BAPO often trains with a reduced effective batch size, offsetting re-evaluation costs.

---

## 3. **Generalizability to other RL algorithms**
**Raised by:** Reviewer SdcC questioned whether BAPO's framework generalizes beyond GRPO to algorithms like PPO.

**Addressed by:** We extended BAPO to PPO (**BA-PPO**) and conducted experiments in **Appendix A.7 (Figure 16)**. Results show BA-PPO significantly outperforms standard PPO, confirming our batch adaptation paradigm is algorithm-agnostic.

---

## 4. **Clarification of adaptive threshold mechanism and reproducibility**
**Raised by:** Reviewers Upr9 and x1w9 noted the adaptive threshold mechanism was unclear and requested explicit formulas for reproducibility.

**Addressed by:** We explicitly provided the linear mapping formulas for adaptive thresholds $c_1$ and $c_2$ in **Section 3.3**:
$$
c_1 = r_{\text{avg}} \cdot (t_{\text{max}}^1 - t_{\text{min}}^1) + t_{\text{min}}^1
$$
$$
c_2 = r_{\text{avg}} \cdot (t_{\text{max}}^2 - t_{\text{min}}^2) + t_{\text{min}}^2
$$
where $t_{\text{min/max}}^1, t_{\text{min/max}}^2$ are boundary hyperparameters detailed in the Appendix. This ensures full reproducibility and clarifies the curriculum-based adjustment.

---

## 5. **Stability of historical high-quality samples**
**Raised by:** Reviewer Upr9 questioned the assumption that historical high-quality samples remain valid under the updated policy.

**Addressed by:** We added **Figure 15 (Appendix A.6)**, which visualizes the accuracy transition matrix of samples throughout training. The results empirically validate that high-quality samples rarely degrade, supporting their reuse without frequent re-evaluation.

---

## 6. **Comparison with prior replay-based methods**
**Raised by:** Reviewer x1w9 noted that techniques like replay buffers have precedents and asked how BAPO differs specifically for LLM reasoning.

**Addressed by:** We clarified BAPO's distinct contributions for LLM RLVR:
- Uses verifiable binary rewards to stratify sample difficulty
- Enforces a delayed refresh mechanism to bound policy divergence
- Structures the batch as $\mathcal{B} = \mathcal{X}_1 + \mathcal{X}_2 + \mathcal{X}_3$ to balance stability and progress

The **"Mini-test"** validates that performance gains are architectural, not heuristic.

---

Overall, we believe that the additional experiments, theoretical grounding, and clarifications have effectively addressed the reviewers’ concerns and strengthened the contributions of our work.

We thank the reviewers again for their insightful comments, which have greatly improved the paper.

---

### Meta-Review · Area_Chair_3uCr · 2025-12-20

**Summary:**

The paper introduces Batch Adaptation Policy Optimization (BAPO), an off-policy Reinforcement Learning with Verifiable Rewards (RLVR) framework designed to enhance data efficiency in LLM post-training. BAPO addresses the limitations of traditional on-policy frameworks, e.g. experience waste and reward homogeneity, by dynamically constructing training batch. The framework utilize three distinct sample types: (1) online generated samples with a zero-advantage filter, (2) historically difficult samples, and (3) historical high-quality samples. Extensive experiments across mathematics, planning, and visual reasoning tasks show that BAPO achieves performance improvement over GRPO.

The initial reviews recognized the work for its well-motivated off-policy framework and comprehensive experiments across multiple domains. However, reviewers initially raised concerns regarding the method's reliance on heuristic hyperparameters, its computational overhead due to re-evaluation, and its generalizability to other RL algorithms. During the rebuttal phase, the authors addressed most concerns, which I will summarize in the following part. After checking the updated manuscript, I find these revisions are well reflected. Note that such a method requires a complicated balance between three criteria of samples, I suggest a poster acceptance and encourage the author include the new results in the final version.

**Reviewer Concerns:**

### Addressed Concerns

(1) Regarding the novelty and hyperparameter Sensitivity, Reviewers g1rk and Upr9 questioned if the performance relied on specific "empirical tricks" or hyperparameter tuning. The authors addressed this with a "Mini-test" using only raw theoretical principles.

(2) Regarding computational overhead or efficiency, Reviewers g1rk and SdcC expressed concern over the cost of re-evaluating historical samples. The authors clarified that BAPO uses a FIFO mechanism to limit buffer size.

(3) Regarding the stability of resuing samples, Reviewer Upr9 doubted the validity of reusing historical high-quality data under an updated policy. The authors provided an accuracy transition matrix to show the connection between sample quality and policy performance.

**Reviewer Scores:**

Most reviewers were positive following the revisions. (1)Reviewer x1w9 raised their score from 4 to 6 after the authors addressed concerns regarding reproducibility and comparison with prior methods. (2) Though Reviewer g1rK initially gave a 4, the authors' detailed technical clarifications on computational efficiency and the mini-test are expected to positively influence the final assessment.

---

### Decision · Program_Chairs · 2026-01-26

Accept (Poster)